# A hybrid data-driven approach to analyze the drivers of lake level dynamics

Márk Somogyvári[1], Dieter Scherer[2], Frederik Bart[2], Ute Fehrenbach[2], Akpona Okujeni[1,3] and Tobias Krueger[1]

[1]Integrative Research Institute on Transformations of Human-Environment Systems (IRI THESys) & Geography Department, Humboldt-Universität zu Berlin, Berlin, 12489, Germany
[2]Chair of Climatology, Institute of Ecology, Technische Universität Berlin, Berlin, 12165, Germany
[3]Earth Observation Lab, Geography Department, Humboldt-Universität zu Berlin, Berlin, 12489, Germany

*Correspondence to*: Márk Somogyvári (mark.somogyvari@hu-berlin.de)

**Abstract.** Lakes are directly exposed to climate variations, as their recharge processes are driven by precipitation and evapotranspiration, and they are also affected by groundwater trends, changing ecosystems and changing water use.

In this study, we present a downward model development approach that uses models of increasing complexity to identify and quantify the dependence of lake level variations on climatic and other factors. The presented methodology uses high-resolution gridded weather data inputs that were obtained from dynamically downscaled ERA5 reanalysis data. Previously missing fluxes and previously unknown turning points in the system behavior are identified via a water balance model. The detailed lake level response to weather events is analyzed by calibrating data-driven models over different segments of the data timeseries. Changes in lake level dynamics are then inferred from the parameters and simulations of these models.

The methodology is developed and presented on the example of the Groß Glienicker Lake, a groundwater-fed lake in eastern Germany, that has been experiencing increasing water loss in the last half century. We show that lake dynamics were mainly controlled by climatic variations in this period, with two systematically different phases in behavior. The increasing water loss during the last decade, however, cannot be accounted for by climate change. Our analysis suggests that this alteration is caused by the combination of regional groundwater decline and vegetation growth in the catchment area, with some additional impact from changes in the local rainwater infrastructure.

## 1   Introduction

One of the most visible effects of climate change in recent years has been the decline of surface water levels, especially in lakes (Woolway et al., 2020). However, not all lakes react to changes in climate in the same way; some are more exposed to climate variations, while others are more exposed to anthropogenic effects (Mason et al., 1994). Understanding the drivers and their importance on lake level dynamics is thus essential for the development of mitigation measures or conservation strategies. The response of lake levels to changing meteorological conditions has been a focus of research for many decades, but in recent years an increased interest in water availability has broadened this research topic with more and more cases waiting for practical solutions (Kebede et al., 2006; Schulz et al., 2020; Getachew et al., 2021; Woolway et al., 2020). This broadened interest often comes with the challenge of limited data availability, especially in remote areas or at the beginning of research campaigns (Woolway et al., 2020; Altunkaynak, 2007; Solomatine and Ostfeld, 2008). Hence a practical approach that can work in such conditions is needed.

We here revisit the downward model development approach of (Sivapalan et al., 2003) as a way of tailoring hydrological models to the data availability. Downward model development starts from large scale, low complexity models and then progresses to the smaller-scale processes (Hrachowitz and Clark, 2017). We demonstrate that such downward model

development approach is suitable for lake system understanding, with the goal of identifying the key drivers of lake level dynamics. In our case, these drivers may be climatic variations, changes in natural water fluxes due to land cover changes or groundwater trends, but also changes in water use and water infrastructure. We propose a hybrid data-driven methodology, where the system understanding gained at a specific level of model complexity is used for the design of the higher-complexity models. We start with a water balance model, which then informs the development of a linear regression model operating on a higher temporal resolution. The development may then continue with nonlinear models (e.g. artificial neural networks) or with the higher temporal resolution or spatially distributed models of the catchment. The downward modeling can also fit organically into the development of process-based models, as will be shown.

Water balance modeling, as in our case, is often an initial step in understanding hydrological systems. Water balance models do not require a complete system understanding to function properly (Xu and Singh, 1998), but can very well capture the macro-scale behavior of the system based on a set of in- and outfluxes. On a monthly timescale, these models only need a handful of hydrological variables, hence will often work in limited knowledge cases.

(Mason et al., 1994) used water balance modeling to simulate the responses of the largest closed lakes around the globe and showed that lakes act as natural lowpass filters over any sudden variation in aridity. (Crapper et al., 1996) used such models to predict future levels of the Goran lake in Australia by taking the cumulative sum of the predicted storage change from the model. (Kebede et al., 2006) used a monthly water balance model at Lake Tana and identified that the main driver of lake level change was the variation in rainfall and not human-induced activities. (Schulz et al., 2020) showed that variations in the levels of Lake Urmia were mainly driven by climate, while local agricultural water extraction had little effect on the overall trend. However, the authors also showed that even without affecting the trend or even some dynamic variations, the abstractions weakened the resilience of the lake to climatic changes and the lake levels could be stabilized by limiting abstraction rates.

The main issue of water balance modeling comes from its highly simplified nature; while these models are robust and easy to model with, they are very general and can overlook some of the details that could be relevant for the hydraulic system. One such issue is how to handle any time-lag between the inputs and the lake level changes. This issue is well known, (Langbein, 1961) already suggested to incorporate the time lag with geometric weight functions into a water balance model. However, most studies tend to overlook this issue by simply modeling on coarser timescales (e.g. monthly, yearly).

At the other end of the complexity spectrum, process-based models are constructed via simulating the individual hydrological (physical) processes that affect the lake dynamics, including spatially and temporally differentiated inflows and outflows, lake bathymetry, weather effects, thermal or chemical forcing (Beletsky et al., 2013; Laval et al., 2003; Valipour et al., 2023). Process-based models can resolve the behavior of the lakes at greater spatial and temporal resolution and can help to study and predict the hydrological evolution of lakes even under complex environmental conditions. Lake models can be expanded to include further physical and biochemical processes such as water quality; hence their application range is very broad.

Lake Erie, for example, has been subject to extensive modeling work to support adaptive management (Arhonditsis et al., 2019). (Getachew et al., 2021) combined water balance and process-based modeling in a prediction framework for lake levels at Lake Tana. The process modeling of these studies focused on the recharge dynamics using the Soil Water Assessment Tool (SWAT), but there is also extensive literature using groundwater modeling software such as MODFLOW, that can better handle lake-groundwater interactions (Lu et al., 2022; Dehghanipour et al., 2019). The downside of process-based models is their time-intensive setup, their large number of parameters requiring extensive data and their large computational costs. This is critical in situations where available data and prior knowledge are limited. In the absence of comprehensive data to run and parameterize process-based models, their theoretical superiority over simple water balance models vanishes.

As models of intermediate complexity, data-driven models are based on readily available observations of the investigated system, while the internal system mechanics are approximated using statistical methods. The underlying system behavior is thus approximated from the mathematical relations between the system input and output data (Souza et al., 2016).

In hydrology, data-driven methods are typically used for prediction or management, for which they are frequently embedded into a system dynamics model framework that goes beyond natural science hydrology (Hassanzadeh et al., 2012; Alifujiang et al., 2017). The term data-driven modeling is often used as an overarching term for a wide variety of novel machine learning methods (Zhu et al., 2020a; Elshorbagy et al., 2010a), but usually excludes methods such as time series analysis or regression that are also data-driven by design.

Time-series analyses methods often only consider the lake level timeseries themselves, predicting them based on their own past values. (Şen et al., 2000) analyzed the time-series of the Van Lake, Turkey with linear and non-linear trends, and combined them with a Markov model to predict future lake levels. (Ebtehaj et al., 2019) based a linear prediction on the spectral decomposition of timeseries. Multiple studies showed the applicability of Autoregressive Integrated Moving Average (ARIMA) models in the context of lake water management. ARIMA models use linear regression combined with moving averages and are suitable for short term time series predictions. Hence the approach is very popular for prediction applications in hydrology (Ghashghaie and Nozari, 2018; Irvine and Eberhardt, 1992; Montanari et al., 1997). Simple time-series approaches, however, are limited as they do not use any weather forcing as input as pointed out by (Kakahaji et al., 2013). (Kakahaji et al., 2013) compared multiple prediction methods on the Urmia lake dataset, including water balance modeling, linear predictor models and different machine learning approaches (multi-layer perceptron and fuzzy networks). The authors concluded that in data-scarce scenarios linear approaches are preferred, while non-linear machine learning methods could only outperform them when properly trained.

Linear regression has been widely used to model the responses of hydrological systems to rainfall (Clarke, 1973; Tasker, 1980). Linear regression assumes a linear relationship between the model input and output, with the linear coefficients calibrated based on the misfit of the model (usually by the ordinary least squares method). As this is an easy to use and robust methodology, it has been the standard data-driven approach in geosciences for decades. Linear models are usually fitted deterministically, but they are also suitable to be implemented within Bayesian frameworks for sensitivity analysis and uncertainty quantification (Kroll and Song, 2013). In more recent studies, linear regression is still widely used as the reliable baseline to compare other more advanced methodologies. For example, (Elshorbagy et al., 2010a, 2010b) compared the predictive capabilities of six different methods using linear models as a baseline. Linear models were similarly used by several studies to show the advantages of machine learning methods (Heuvelmans et al., 2006; Sahoo and Jha, 2013).

Machine learning applications are gaining increasing popularity in hydrological practice, for example for rainfall-runoff modeling (Kratzert et al., 2019; Sahoo et al., 2019; Klotz et al., 2020), water resources management (Oyebode and Stretch, 2019), drought prediction (Li et al., 2021) or lake level prediction (Kisi et al., 2012; Demir and Yaseen, 2023). Machine learning methods provide black-box solutions with a non-linear internal mathematical structure. They can be used as predictors based on the lake level variation data only (Zhu et al., 2020b), or to predict dynamics based on forcing data (Páliz Larrea et al., 2021). Machine learning models typically have a very large number of parameters. Proper calibration of parameters requires large datasets, which again limits the applicability of these approaches. An even larger issue regarding the context of our study is the black-box nature of machine learning methods: it is very challenging to analyze individual processes when the black-box method is designed to mimic the overall behavior of the system (Amy McGovern et al., 2019). This is also true for most other data-driven approaches, as they are designed for prediction rather than understanding, but models of less complexity, e.g. regression-type models, could still be analyzed with relative ease.

In this paper we present a case of limited prior knowledge where a process-based model cannot yet be set up with the required level of confidence, although predictions of lake level change and an assessment of potential drivers are increasingly demanded by policymakers and stakeholders. We use the case of the Groß Glienicker Lake, a groundwater-fed lake at the outskirts of Berlin, Germany, that has experienced drastic water losses over the last decades. This loss is not systematically observed in all

lakes of the region (Lischeid et al., 2021), hence further drivers beyond climatic changes need to be examined, e.g. water infrastructure and land use changes.

We follow the downward model development approach as follows: by a monthly water balance analysis, we identify and quantify missing water fluxes in the hydraulic system, and use it as a baseline to identify any turning points and changes over the investigated period. This informs a daily data-driven linear model that can unfold the lake level responses to specific events

in more detail. By identifying the main drivers of the lake level dynamics and system changes, our study will support the development of a future process-based model, while the results can already be used in local water management initiatives.

## 2    Methodology

In this study, we propose a top-down model development approach (Sivapalan and Young, 2005; Hrachowitz and Clark, 2017)

to understand the lake level dynamics, starting from simple water balance models to more complex data-driven approaches, with an outlook to what we can learn from these for even more complex process-based modeling. We propose a hybrid data-driven modeling framework, consisting of the following steps:

1. Monthly water balance model to quantify fluxes
2. Identify main turning points in the system using the water balance residuals

3. Daily linear regression models between the turning points
4. Model response analysis to isolated weather forcings
5. Further analysis of step 3 and 4 with non-linear approaches of increasing complexity (if needed)
6. Triangulation of findings using independent data

Our proposed methodology requires the development of multiple models. First, a water balance model is created on the

monthly scale, that helps quantifying the fluxes of the hydraulic system, and helps identifying any major turning points during the investigation periods. The evolution of the water balance residuals can indicate systematic changes, like increase in outflow from the catchment.

Next, daily-scale data-driven models calibrated over the periods between the turning points are compared in order to analyze the differences in their lake level response. We start this analysis using linear regression models, due to their simplicity and

transparency. The model responses to the different weather forcings (precipitation, evapotranspiration) are compared separately as well, to understand the system in detail.

If linear models cannot capture the system behavior sufficiently, we propose to increase the model complexity using non-linear models, such as artificial neural networks. In our study, the linear approach provided good fits, and enough insight for system understanding based on the available data–- the fact the system seems to behave linearly is in itself an interesting result. As

the last step, the findings are validated against independent information. In the following, first we present the meteorological forcing data, and the way it was obtained. Then, we present the methodologies of the water balance and linear regression approaches.

### 2.1  Forcing meteorological data

The proposed methodology relies on local meteorological data from the investigated lake catchment, which is achieved by using data from the second version (v2) of the Central European Refined analysis (CER) (Jänicke et al., 2017), a gridded meteorological dataset for Central Europe with focus on the region of Berlin-Brandenburg. As its predecessor CER v1, the CER v2 dataset has been produced by an observation-based model approach. Global ERA5 reanalysis data has been dynamically downscaled using the Weather Research & Forecasting (WRF) model, and validated against 211 weather stations.

The methodologically has been comprehensively described and successfully applied in different regions of the world, for instance in High Asia (Wang et al., 2021; Maussion et al., 2014). The CER v2 data set covers the time period from 1980 to 2022 (with continuous update of most recent years) using a convection-resolving approach at the highest spatial resolution of 2 km horizontal grid spacing. Data from 2002 to 2022 have been used in this study since the CER v1 data set, which dates back only to 2002, was used to test the robustness of our methodology.

There are two advantages of using a dynamically downscaled gridded dataset instead of relying on interpolated station data. First, such an approach provides an estimate of actual evapotranspiration for each grid point using land cover, vegetation and soil data, and dynamic data on soil moisture, while station-based observations are typically restricted to potential evapotranspiration (lysimeters or eddy flux towers would be available at only very few locations). Second, this approach explicitly takes into account meso-scale heterogeneity of weather systems, which is of particular importance for precipitation and actual evapotranspiration with high variability at spatial scales of a few kilometers or less. When we tested our lake models using weather station data, we were unable to obtain the same model fit qualities as with the CER v2 dataset. The largest differences happened after extreme rainfall events, where due to the spatial variation the recorded amount of rainfall could differ a lot from the rainfall at other locations. Because summer storms have a strong impact on the lake levels, we could not close the water balance models only using weather station data.

## 2.2 Water balance modeling

In groundwater fed lake systems without any surface water connections, the lake level dynamics will be mainly dependent on the inflow from the groundwater. This flow is controlled by the groundwater level – lake level relation. Hence, the groundwater dynamics and lake level dynamics are strongly related, and the lake level changes can be used as an indicator for the groundwater level changes in the catchment. The water balance equation for a groundwater fed lake system can be formulated as:

$$\Delta S_{lake}(t) = P_{lake}(t) - E_{A,lake}(t) + F_{in}(t) - F_{out}(t) + \epsilon$$

( 1 )

where $\Delta S_{lake}$ is the change in lake water storage, $P_{lake}$ is the total precipitation over the lake and $E_{A,lake}$ is the total lake evaporation. $F_{in}$ and $F_{out}$ are in this case the subsurface in- and outflow of water to the lake, which can be combined into the net subsurface water inflow ($\Delta F$). The final term $\epsilon$ explains any remaining errors and uncertainties in the data. If there were any surface water connections to the lake, an extra net surface water inflow would have to be accounted for. All terms in eq. 1 are expressed in units of volumes over time (e.g. in m$^3$/day or m$^3$/month), with fluxes integrated over the lake surface area.

Precipitation ($P_{catchment}$) and actual evapotranspiration ($ET_{A,catchment}$) over the (subsurface) catchment area, and not just the lake, strongly influence subsurface flow processes that feed the lake. However, these effects show some time delay. Therefore, the water balance equation for the catchment reads as:

$$\Delta S(t) = \int_{t-\tau^*}^{t} P_{catchment}(\tau) \, d\tau - \int_{t-\tau^*}^{t} ET_{A,catchment}(\tau) \, d\tau + \Delta F`(t) + \epsilon$$

( 2 )

Here, the first integral sums precipitation over the catchment back over time until a precipitation event ceases to cause an inflow into the lake at time $t$, while the second integral does the same for actual evapotranspiration. We denote this time interval with $\tau^*$ which we can also call the hydraulic memory of the system, which is sometimes called lake response time in the literature (Mason et al., 1994; Gong et al., 2015). Here, $\Delta S$ denotes the change in storage over the whole catchment, which is mainly the change in groundwater storage. With this assumption, storage changes in the unsaturated zone are neglected. In the model, this time represents the time water spends travelling through the unsaturated zone, and then the pressure impulse traveling through the system. The hydraulic memory of the system is estimated from the observed data.

In eq. 2, precipitation over the lake is included in the precipitation of the catchment and evaporation is included in the catchment evapotranspiration term. The modified water balance equation leaves $\Delta F$ ` to account for any remaining net subsurface inflow unaffected by climatic forcing, for instance water abstractions in which case $\Delta F$ ` would be negative, or diverging regional groundwater flows. If these flows are approximately constant over the investigated time period, they will not appreciably affect the lake level dynamics.

Considering the discrete nature of daily input data, the integrals can be substituted by sums:

$$\Delta S(t) = \sum_{i=1}^{\tau^*} P_{catchment}(t-i) - \sum_{i=1}^{\tau^*} ET_{A,catchment}(t-i) + \Delta F` + \epsilon$$

( 3 )

The complete water balance then can be used to estimate the changes in catchment storage. To use such model for the lake level dynamics, the catchment storage change ($\Delta S$) needs to be converted to lake level change ($\Delta z$). Lake level change can be estimated from lake storage change using a bathymetric model, but this approach is not suitable for catchment storage. Hence, we used an assumption that lake level changes are linearly related to catchment storage changes. We based this assumption on the fact that lake level changes are relatively small compared to the scale of the catchment, and the catchment geometries are simple in a lowland, sedimentary geological setting. The catchment storage change - lake level change relation reads as:

$$\Delta z(t) = \alpha \Delta S(t) + \beta$$

( 4 )

The slope ($\alpha$) and intercept ($\beta$) can be estimated by optimizing the fit between the observed and modelled lake level changes. In simpler hydrological systems (such as our case) $\alpha$ could equal 1, and $\beta$ zero. The units of $\alpha$ is one over area ($1/m^2$) to account for the conversion from change in volume to change in depth, the units of $\beta$ is the same depth over time as for $\Delta z$ (e.g. m/day or m/month).

The water balance equation is closed by calibrating the $\Delta F$ value aiming to ensure the best fit between the observed and modelled lake level changes. To characterize the fit quality, we use the $R^2$ score metrics, the coefficient of determination. The calibration is done via a simple grid search, by testing a series of values with reasonably small intervals between them.

After the water balance equation is closed the remaining residuals ($\epsilon$) are analyzed to identify any systematic trends or turning points in the system. Turning points can be identified where as the starting point of a continuous increase or decrease in the residual values (see Fig. 5 for example). To quantify these effects, transient fluxes can be introduced to the water balance, with constant values within certain time intervals. The calibration of such fluxes is done similarly, using a grid search.

## 2.3 Data-driven modeling

Data-driven models use the statistical relationship between the model input data and the observed outputs. Based on the modified discrete water balance equation (eq.3), we can frame the general modeling problem as:

$$\Delta z(t) = f\big(P(t-\tau^*), \dots P(t), ET_A(t-\tau^*), \dots ET_A(t)\big) + \epsilon$$

( 5 )

This means that we are looking for the functional relationship between the meteorological input data (fluxes in the units of mm/day considering a daily timescale) and the observed lake level changes (in m/day). This equation can be amended by additional input data, like data on water abstraction (if such data is available).

The simplest function that can be used in this model is a linear function, which would read as:

$$\Delta z(t) = a + b_{P,-\tau}P(t-\tau^*) + b_{P,-\tau+1}P(t-\tau^*+1) + \cdots$$
$$+ b_{P,0}P(t) + b_{ET,-\tau}ET(t-\tau^*) + \cdots + b_{ET,0}ET(t) + \epsilon$$

( 6 )

Where $a$ is the intercept of the linear function, $b_{P,-\tau+i}$ are the linear coefficients for precipitation for the timesteps $\tau + i$ in the past, $b_{ET,-\tau+i}$ are the respective coefficients for actual evapotranspiration. Although this is a relatively simple formula, the function could have a high dimensionality, which increases with the memory $\tau^*$ and the number of input features $F$ ($f: \mathbb{R}^{F\tau^*} \to \mathbb{R}$). This often referred to as a multilinear problem in the literature (Sahoo and Jha, 2013).

The linear model can be split into two sub-models, to investigate the individual responses to precipitation and evapotranspiration respectively:

$$\Delta z_P(t) = b_{P,-\tau}P(t - \tau^*) + \cdots + b_{P,0}P(t)$$

$$( 7 )$$

$$\Delta z_{ET}(t) = b_{ET,-\tau}ET(t - \tau^*) + \cdots + b_{ET,0}ET(t)$$

$$( 8 )$$

Optionally, input data might be filtered prior to the analysis. In this study we used Butterworth filters from the scipy.signal python package. For the autocorrelation analysis in section 4.1, a bandstop filter was used, that removes the 365 days period signal from the lake level data. For the plots of the linear regression analysis (Figure 6,7,8 and 9), a lowpass filter was used over the lake level data, with a cutoff frequency at 20 days. This was necessary for the visualization in Fig. 7, where the higher frequency components would appear as noise over the coefficients. A comparison plots of the filtered and unfiltered data is shown in supplementary figure S1.

To make the linear model coefficients more comparable for the different input types, the input data may be standardized as well. Standardization rescales the input timeseries to have a zero mean and a standard deviation of 1. Standardization may also be necessary for non-linear machine learning approaches.

The linear model is fitted using ordinary least squares regression, from the scikit-learn python library (Pedregosa et al., 2011). The method minimizes the sum of squared errors between observed and simulated data using an explicit formula of a projection matrix.

The lake response time/system memory can be estimated in multiple ways, in this study we used two separate methods. First the k-lag autocorrelation of the lake level data was calculated.

$$AC(\tau) = corr(z(x), z(x - \tau))$$

$$( 9 )$$

The k-lag autocorrelation shows the time dependence of the lake level data, and give a good indication of the ideal memory timeframe for the modeling (Seeboonruang, 2015). A second approach, which is often used in rainfall runoff models, is fitting and evaluating a series of linear regression models, with different memory windows. With this approach the fit qualities of the different models are compared to identify the lake response time.

### 3    Study site and data

We applied this method to the Groß Glienicker Lake, a groundwater fed lake system at the border of Berlin, and the federal state of Brandenburg in Germany. Like several lakes in the region, the Groß Glienicker Lake has shown a drastic loss in lake levels over the last half century, with an increasing rate over the last decade (Fig. 1b).

The lake catchment delineated from topographic data spans over 33 km², mainly consisting of forest (20 km²), cropland (6 km²) and urban area (4 km²). The main recharge area is the heathland west of the lake dominated by sandy soils. The regional

groundwater flow direction points southeast, toward the river Havel, a major tributary of the river Elbe, but due to the lack of groundwater wells and low gradients, the exact groundwater flowsystem is currently not known.

Two lakes are located within the catchment, the Sacrower Lake and the Groß Glienicker Lake. The latter has been chosen for this study because it is a focus of the local concerns. The lake nevertheless is representative of declining lake levels that are widespread in the Berlin-Brandenburg region and beyond. Both lakes are groundwater fed, with no active surface water connections, similarly to many lakes in the region (Lischeid, 2021). A connection between the two lakes used to exist, but it has been closed in 1996 due to the declining levels in both lakes.

The Groß Glienicker Lake has been extensively studied from a hydrochemical point of view because between 1970 and 1990 a large amount of untreated sewage was regularly discharged into the lake from a nearby army base leading to eutrophication. To mitigate the effects of this pollution, a restoration campaign started in the early 1990s, which is well documented (Wolter, 2010; Kleeberg et al., 2012; Heinrich et al., 2022). There are, however, only a limited number of studies that focused on lake level dynamics, although the continuous decrease in lake levels has been a concern for the local communities and authorities for a while.

The lake is located on the administrative boundary of the German federal states of Berlin and Brandenburg. This makes the accessibility of infrastructural and water management data complicated. From the Berlin side, data regarding water supply, wastewater management and canalization maps are available on the city webpage for multiple years. From the Brandenburg side, geological and limited information on the rainwater infrastructure (e.g. manhole cover locations) is available. The lake levels are monitored by an automatic measurement station at the south side of the lake operated by the city of Berlin. Daily lake level data since January 1970 are openly available on the website Wasserportal Berlin. The hypsographic curve based on the bathymetric model of the lake shows a linear relation between the lake volumes and lake levels (Jahn and Witt, 2002: p66), but note that this relation cannot be used to link the lake levels to the catchment storage.

The catchment shown in Fig. 1a, was delineated using surface topography data, hence it represents the surface catchment. This is used throughout the analysis, instead of the unknown groundwater catchment. Due to the integrative nature of our analysis, and the focus on the lake level dynamics, this is not a big issue, but it could cause uncertainties when the exact quantification of the hydrological fluxes is needed (see section 4.2). The catchment is located in a lowland area, with an elevation range of 40 meters. The unsaturated zone depth is up to 15 meters (Geoportal Brandenburg - Detailansichtdienst, 2024).

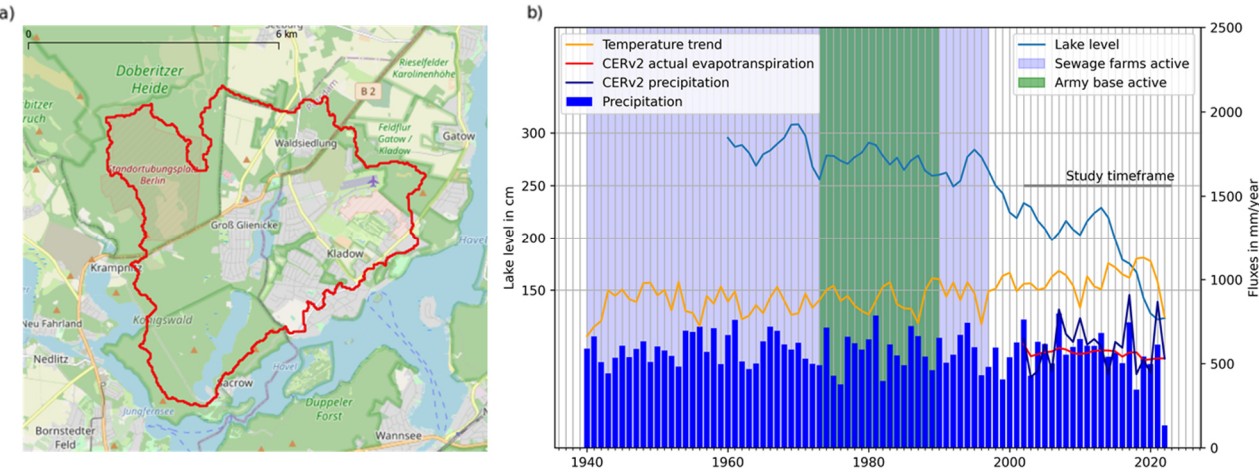

**Figure 1: a) Catchment of the Groß Glienicker Lake (© OpenStreetMap contributors 2023. Distributed under the Open Data Commons Open Database License (ODbL) v1.0.), b) Overview of lake level changes of Groß Glienicker Lake together with concurrent timeseries of temperature and precipitation. Key events potentially impacting the lake system are shown as background colors.**

The Central European Refined analysis (Jänicke et al., 2017) provides atmospheric data for the investigation region Berlin-Brandenburg on a spatial grid of 2 km, at a temporal resolution of hours. In this study, we used the daily aggregated data of precipitation and actual evapotranspiration, integrated over the catchment area of the lake. Actual evapotranspiration is calculated from atmospheric parameters using static land use data. The exact land use composition of the catchment was estimated from the 2015 remote sensing based land cover analysis of (Pflugmacher et al., 2019). Figure 2 shows an overview of the CER v2 data over the study timeframe of 2002-2023.

## 4    Results

In Figure 1 air temperature shows a very apparent increasing trend over the last decades. Precipitation does not show any long-term trends, only shorter-term variations. This is in line with the climate analysis of the German Weather Service, which forecasts a slow increase in precipitation in the Brandenburg region (DWD-Deutscher Wetterdienst, 2019). It is also stated, however, that extreme events are becoming more frequent and have provided a larger fraction of the annual precipitation in later years.

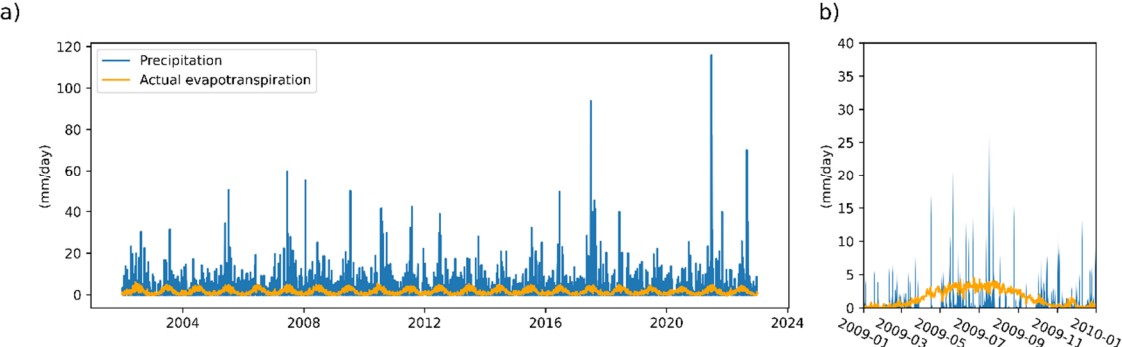

**Figure 2: Weather forcing input data from the CER v2 dataset: a) investigation period, b) example year of 2009.**

Figure 2 shows how summer periods are dominated by the extreme rainfall events in the catchment. These events are isolated by drought periods, as seen during the example year of 2009 plotted in Fig.2b. The data also shows that actual evapotranspiration has a much more periodic and regular behavior, with similar patterns over the years. However, the downscaled actual evapotranspiration data do not show any increasing trend, which one might expect from the increasing air temperatures (Fig. 1b). This however would only happen in energy-limited systems with unlimited water availability (e.g. over open water bodies). While potential evapotranspiration would follow such temperature trend, actual evapotranspiration in water limited regimes does not depend on air temperature.

### 4.1 System memory (lake response time)

The memory of the hydraulic system is estimated by calculating the k-lag autocorrelation of the lake level data. A bandstop filter is used over these data to remove the annual cycle, which dominates the lake level periodicity and could distort the analysis (see supplementary Figure S1 for the filtered timeseries). The result is shown in Fig. 3a.

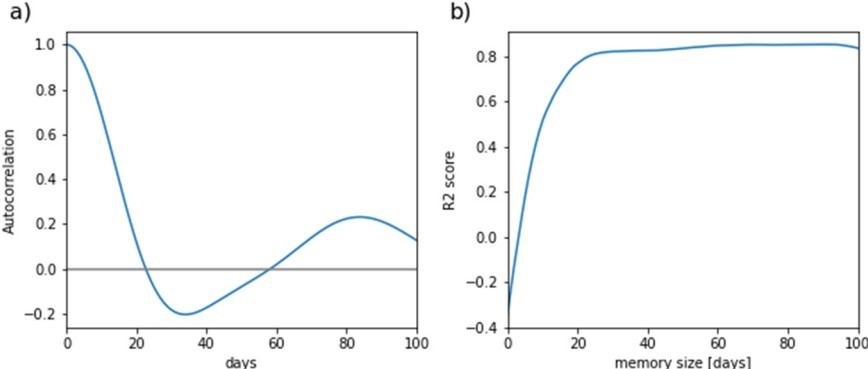

**Figure 3: Lake response time analysis: a) autocorrelation of lake levels, b) fit quality of linear models of the weather forcing-lake level relations (eq. 6) using different system memory timeframes.**

The autocorrelation plot in Fig. 3 shows a rapid decrease, it reaches zero around 20 days, and its minimum value around 30 days. Another method for estimating the optimal memory timeframe in hydrology (mainly in rainfall runoff modeling studies) is to compare linear models with different memory lengths over the same data. This is shown in Fig. 3b, in a range from 1 to 100 days using the $r^2$ metric. The overall picture is very similar to the autocorrelation, with a rapid increase in fit quality until about 20 days. Then the fit quality increase slow down and stays at a high value of 0.8. This large range of optimal fits indicates the robustness and insensitivity of the linear regression method. Based on these analyses, we will use 30 days as the lake response time or hydraulic memory throughout this study.

## 4.2 Water balance

The water balance model is built on a monthly scale (30 days scale) – suggested by the system memory analysis. The monthly precipitation ($P(t_m)$) and actual evapotranspiration ($ET_A(t_m)$) timeseries are generated via summing up the daily values, and the lake level timeseries (which is used for model validation) is averaged to monthly means. The monthly weather values are then compared with the mean lake level of the next month ($\Delta z(t_{m+1})$).

$$\Delta z(t_{m+1}) = P(t_m) - ET_A(t_m) + \Delta F`$$

( 10 )

Eq. 10 shows the used water balance equation. All terms in the equation have units in mm/month.

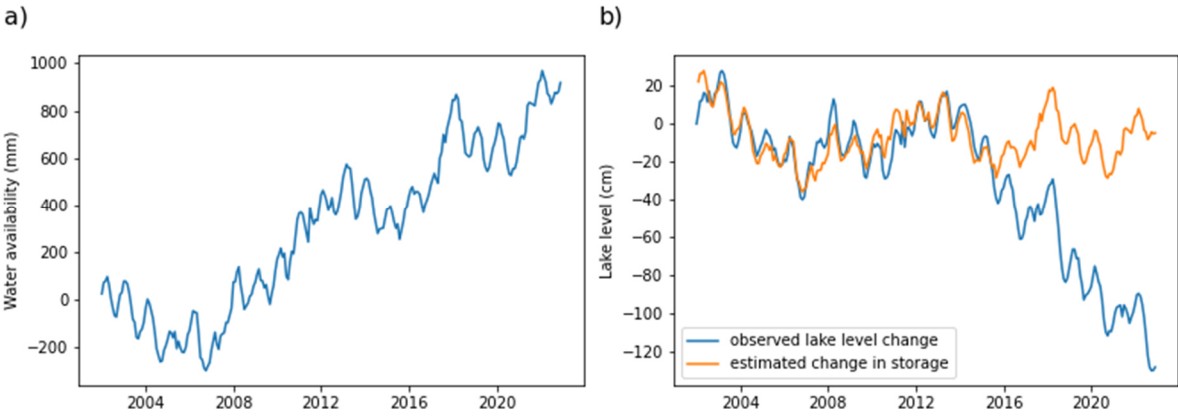

**Figure 4: Water balance modeling: a) cumulative sum of water availability, b) observed and estimated lake levels.**

Figure 4 shows the two main steps of the water balance estimation. First, the water availability is calculated by subtracting actual evapotranspiration from precipitation monthly. By taking the cumulative sum of the water availability, we can see that if only these two processes would affect the lake, the lake levels would increase over the investigation period.

To obtain a more realistic picture of the water level changes an additional loss term needs to be introduced ($\Delta F`$ in eq. 10). In this case, a constant outflow equivalent of 4.5 mm per month was necessary to bring the water balance curve as close to the lake levels as possible. This flux was estimated via a grid search via maximizing the $r^2$ score and can be attributed to the net groundwater outflow of the system.

There is a clear breaking point between the modelled and the observed data around 2015, where the two curves start to diverge from each other. Before this turning point, the obtained fit was maximal at 0.76. This means that until this point in time, 76% of the lake level variations can be explained solely by the variations of the meteorological inputs. After this point in time, the difference between the curves significantly increases, as it is shown by the misfit curve in Fig. 5.

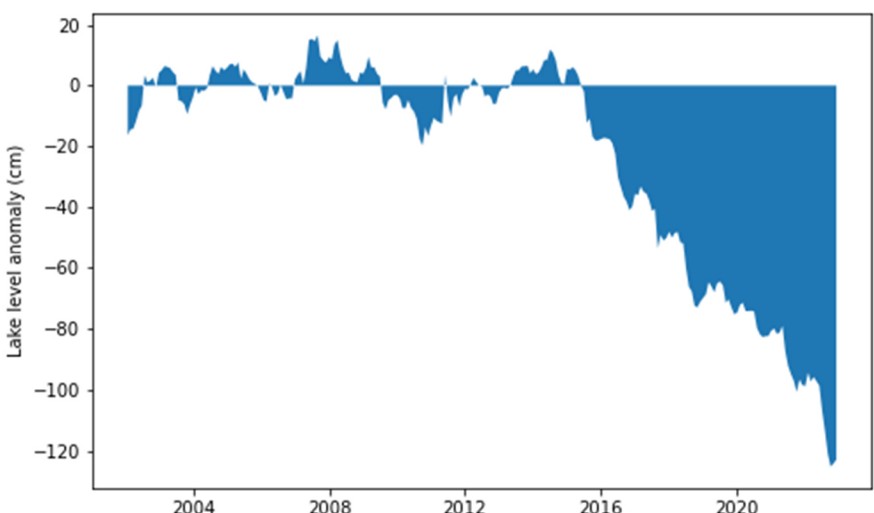

**Figure 5. – Differences between the observations and the water balance model (non-climatic water balance anomaly). At positive values there is surplus water in the lake compared to the model, at negative values the lake shows a water deficit that is unaccounted for by the model.**

It is clear from Fig. 5 that the residual variations cannot be explained by a single missing water balance component, but with some system change. Until 2015, the model is in good agreement with the lake level observations, despite some short-term variations in each direction.

Between 2015 and 2022, the lake levels exhibit a downward trend (Fig. 4), which is not captured by the model. By 2015, we see a systematically widening overestimation of the observed lake levels (Fig. 5). The difference is a rainfall equivalent of 10 mm every month, which given the catchment size is around 4 million m$^3$ yearly. Because this change in the water balance happens very quickly, the time of change is very well identifiable at 2015 which is a strong turning point in the hydrological system.

### 4.3 Linear model

To investigate the changes in the hydrological system in more detail, two data-driven linear models were constructed. The two models are set up identically, both taking daily precipitation and actual evapotranspiration data as input, with a 30-day long memory into the past (eq. 6). The lake level data is filtered using a Butterworth filter with a 20-day cutoff frequency. The only difference is the calibration period used. The model 2004 uses a seven-year period after 2004 for calibration, which is a relatively steady period according to the previous analysis, while the model 2015 uses the last 7 years of the dataset from 2015, after the previously identified turning-point. The calibrated models are then run for the complete available timeseries, adjusting the start of each calibration period to the actual lake levels that day. The results are shown in Fig. 6.

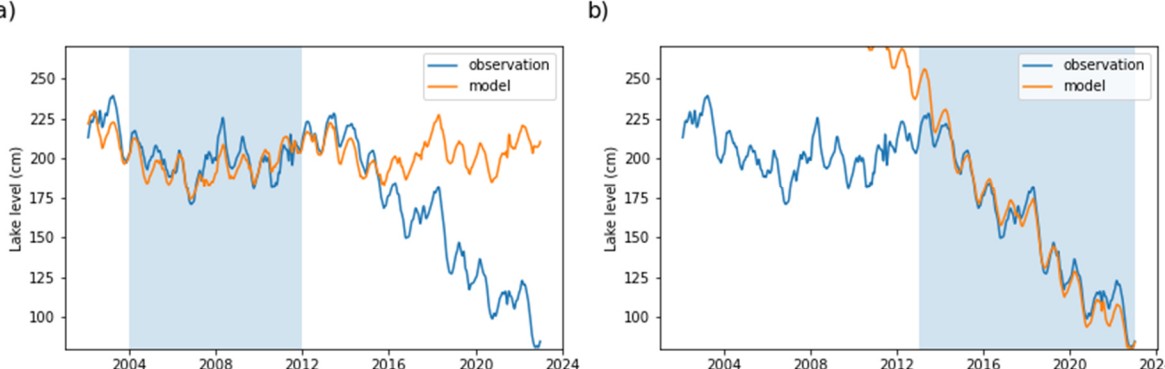

**Figure 6: Daily lake levels modelled with a linear model using different time periods for calibration (marked by blue shading): a) model 2004, b) model 2015.**

The results in Fig. 6 clearly show the different dynamics of the two investigated time periods. The model 2004, trained on the earlier period depicts a very similar behavior to the water balance model. An overall good fit in the first 12 years, with some larger deviations in more extreme years like 2007, which was exceptionally dry. After 2015, the model systematically overestimates the lake level and hence an increasing gap opens between the observed and modelled lake levels. The gap is very similar to the water balance model. As the linear model was calibrated independently from that model, its similar result

provides a validation for the chosen water balance parameters.

    The output of the model 2015 is very much the opposite. It calibrates so as to capture relatively well the final steep decrease of the lake levels (the fit is even better than the first model), but when this modelled trend is extrapolated over the first half of the dataset as well, it overestimates the lake levels.

    These results support the conclusion that the lake system behavior changes systematically around 2015. To diagnose these

changes further, we now look into the calibrated models, their mathematical structure and their response behavior. Note that as these models are purely data driven; any missing process in the data is compensated for by adjusting the coefficients for P and ET.

    To make the estimated effects of the different input features comparable (the effects of different predictors), the input data are first standardized for this analysis. The model coefficients thus give a good indication of the importance of the different inputs

over the model's memory framework. The results for the two investigated time periods are compared in Fig. 7.

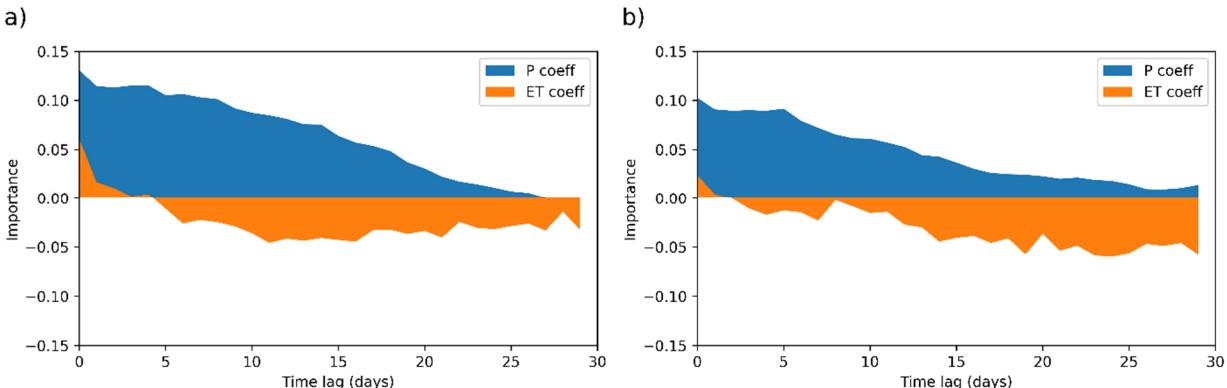

**Figure 7: Model coefficients for precipitation and actual evapotranspiration: a) model 2004, with calibration period of 2004-2011, b) model 2015, with calibration period of 2015-2022.**


    Figure 7 shows the coefficient values of the two models over different time lags. For example, the precipitation coefficient at time lag 5 is the weight with which the precipitation 5 days ago enters the calculation of today's lake level change (see eq. 6). This plot shows that the lake reacts to precipitation and evapotranspiration in a different manner, and that this difference

changes depending on the calibration period due to the hypothesized system changes. The effect of precipitation is detectable
immediately, and days in the past are becoming less and less relevant. In model 2004, the rainfall importance is generally higher than in model 2015, where it decreases rapidly after the first 10 days.

These findings can be explained with the following conceptualization: after rainfall, as rainwater reaches the groundwater table it creates a hydraulic gradient, and the hydraulic signal reaches the lake very rapidly. The impact of the rainfall is still visible a few days later, as some of the water takes more time to seep through the soil. This impact decays over time continuously.
Actual evapotranspiration, on the other hand, has a delayed influence on lake level changes with a variable but on average constant importance of past days after 5-10 days in model 2004 and an increasing importance of past days after 10-15 days in model 2015. The overall importance of actual evapotranspiration is also higher in the second model (34% vs 42%).

To further analyze the model behavior, we created two sub-models according to eq. 7 and 8. Here, we separated the lake level response for the two input features, precipitation and actual evapotranspiration, i.e. simulating the lake level responses that
would result from only precipitation or actual evapotranspiration as predictors. We used non-standardized inputs and outputs for this analysis. This way we can directly compare the differences between the two models in terms of the different effects of the two inputs. This is shown exemplary in Fig. 8.

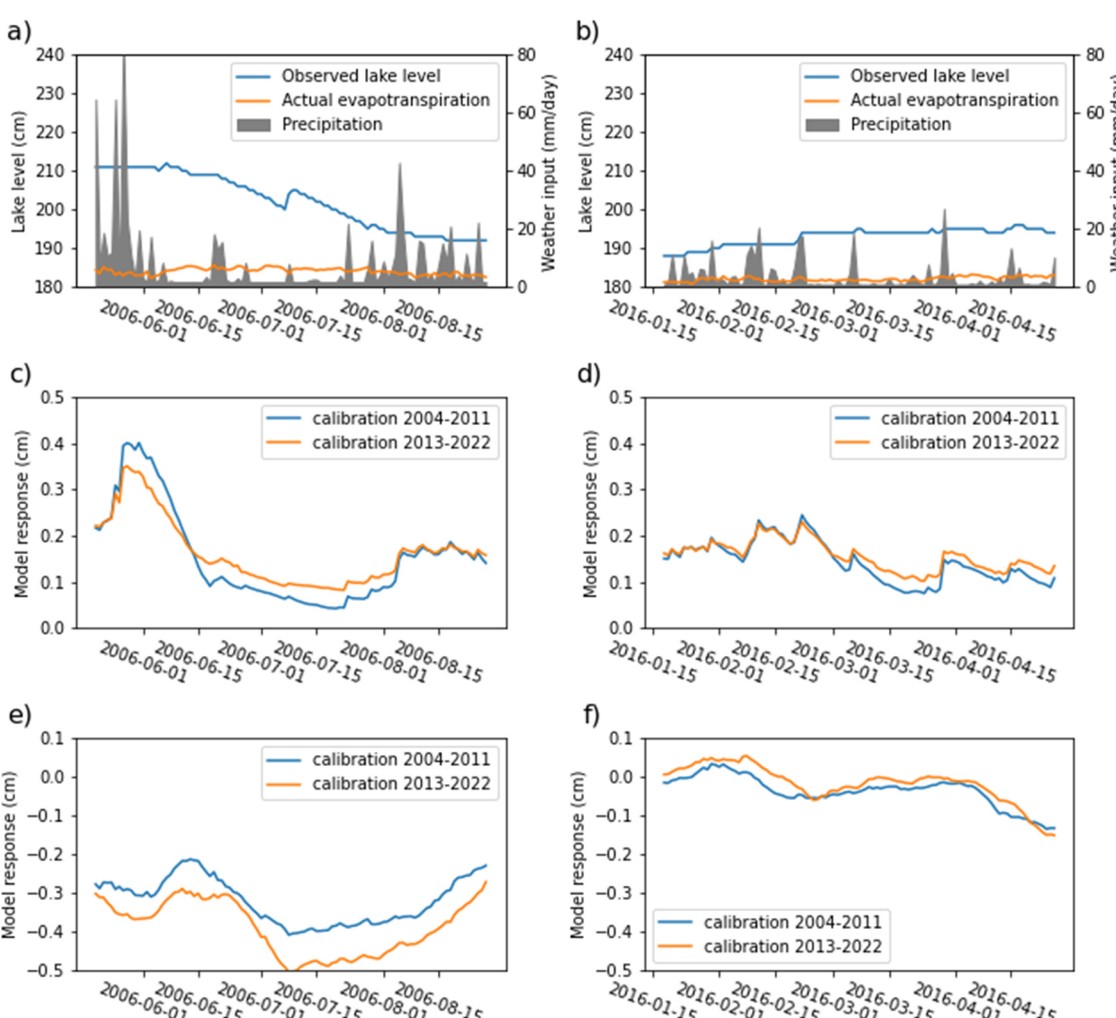

**Figure 8: Modelled hydraulic responses to different weather patterns: a) summer weather input with high intensity storms, b) spring weather input with light rain, c) model response to precipitation in summer, d) model response to precipitation in spring, e) model response to actual evapotranspiration in summer, f) model response to actual evapotranspiration in spring.**

In this plot we zoom into two different parts of the dataset to compare the two models directly in detail, focusing on typical weather events. The first time period in Fig. 8a is the late summer of 2006, which saw many days without precipitation but
high evapotranspiration and single day rainfall events with relatively high amounts of rainfall. During this time period the

model 2015 shows a systematically stronger response to actual evapotranspiration (Fig. 8e), which leads to a larger lake simulated level drop. The offset is not just vertical, there is a time lag of 5-10 days between the two responses (as expected based on the coefficients in Fig. 7).

The precipitation response is a bit more complex: the model 2004 gives a much stronger response to the larger rainfalls, but a
weaker one to the lack of rain (Fig. 8c). This balances out the two curves over this time period – resulting in a similar precipitation response.

The models behave differently during the calmer spring season of 2016 (Fig. 8b). Here both the two evapotranspiration responses (Fig. 8f) and precipitation responses (Fig. 8d) run close to each other with small differences.

Therefore, the general offset between the two models is systematic. Between September and June, the models behave similarly
– this is a period with regular rainfall without many dry days, or extreme rainfall events. The discrepancy in these periods however is usually not that high, hence the two models stay relatively close to each other.

This result shows that the main difference in the system between these two time periods is coming from the difference in evapotranspiration during the summer periods. The seasonality of the model differences is shown in more detail in Fig. 9, where the median of the lake level response differences for the two inputs are plotted over the months of the year.


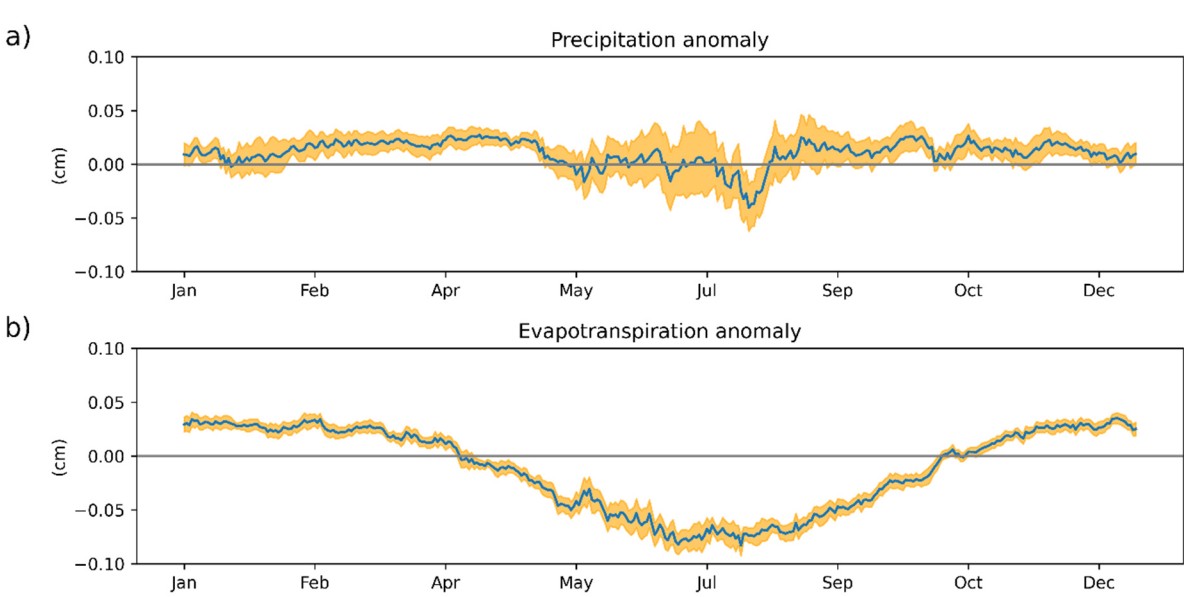

**Figure 9: Yearly dynamics of the model discrepancies (model 2015-model 2004): a) median difference in precipitation response with confidence intervals, b) median difference in evapotranspiration response with confidence intervals.**

The median difference in precipitation response in Fig. 9 shows that over the long run, the differences in rainfall response
between the two models are cancelled out. We can see some small positive anomalies in the spring and fall, but this effect is much smaller than what is visible in evapotranspiration.

Evapotranspiration response also shows a small positive anomaly during these periods, keeping the two models close during the winter. Figure 9b nicely shows that the main difference between the two models originates from the summer evapotranspiration difference. This difference is very consistent over the years, indicated by the narrow confidence interval –
which is not surprising as rainfall shows the bigger variability over the years.

## 5     Discussion

The water balance model shows a 4 million m$^3$ yearly deficit to the climatic water balance since 2015. We take the two linear models as representing the system behavior during the relatively stable period between 2002-2015 and between 2015 and 2022,

respectively. The change in system behavior between these two periods is projected onto differences in responses to precipitation and actual evapotranspiration in these simple models while in reality a number of other processes will be responsible. However, the changes in the responses can still be analyzed to hypothesize about the actual processes at work. In this section we discuss some of these hypotheses.

**5.1 Water management**

A possible factor responsible for the accelerated decrease in lake water levels since 2015 that has been put forward by local stakeholders is an increase in water abstractions at the nearby water supply wells. Our analysis, however, does not support this hypothesis. There was no reported change in abstraction rate of the local waterworks in this period, and an increased abstraction rate would hardly explain the change in the short-term system dynamics – it would appear as a constant shift in water loss instead. Nevertheless, groundwater abstractions could affect the resilience of the lake to climate change effects as was shown by (Schulz et al., 2020) for Lake Urmia. Based on process-based modeling, their study did not find a direct correlation between abstractions and water level variability of Lake Urmia, as the lake could buffer the reduced inflow. However, in forecast scenarios, they achieved higher lake levels with reduced abstraction rates.

Abstractions by local households directly from the lake or from the groundwater could also have an effect, as people tend to use these water sources for gardening. To calculate an upper bound for such private abstractions, we assume that all 5000 residents of Groß Glienicke (on the Brandenburg side of the lake) use 7% for gardening (Schleich and Hillenbrand, 2009) of the 200 l daily average water consumption (OECD averages), which would amount to 26 250 $m^3$ abstraction over the year. This is significantly less, than the estimated water deficit.

Another local water utility is a former sewage farm (Rieselfeld Karolinenhöhe) north of the catchment. Here, large volumes of untreated wastewater were infiltrated into the groundwater system up until 2010. The effects of this facility have been extensively studied (Haacke et al. 2018; Liese et al. 2004) but no direct link has been found between the sewage farm and the lake's catchment as the infiltrated water flew directly into the Havel river. The sewage farm stopped its operation in 2010, well before the identified turning point.

Another infrastructural change that happened in the area was an upgrade of the sewage system. The most notable example is the former British air force base (General Steinhoff Kaserne) east from the lake. Here, an almost 500 000$m^2$ area got connected to the rainwater canalization system between 2012 and 2017, and the new system now leads the collected rainwater to the Havel river, outside the catchment (Döllefeld et al., 2021). Assuming that 90% of this water would have previously reached the groundwater or the lake (an assumption based on the urban evapotranspiration fraction of the CER dataset), this could account for up to 225 000 $m^3$ of the missing fluxes from the catchment.

Dialog with the local community also suggested that this canalization upgrade extended beyond the former airbase and it also might have included the sewage system. Unfortunately, no reports or studies are available, but similarly to the private abstractions, we can estimate the sewage production of the districts around the lake. Estimating 20 000 residents living in the area, with an average sewage production of 120 l per year (Umweltbundesamt, 2023), we arrive at an estimate of 900 000 $m^3$. This could be used as an upper bound for the potential effect of the infrastructure change, if before all of this water had been discharged in the catchment directly. This, however, is most likely a big overestimation, as the area is shown to be connected to the Berlin canalization system in documents from 2012 and before (https://www.berlin.de/umweltatlas/wasser/regen-und-abwasser/2012/literatur/). The upgrade most likely affected the rainwater canalization only, which has a reported average yearly flux of around 100 000 $m^3$.

Still, this amount could have affected the hydrological system in some way (see Fig. 11), but cannot explain the observed misfit. In particular, the lack of infiltrated water in the catchment would not explain the observed seasonality in the misfits (Fig. 9). Further analysis of this issue is limited because the lake is located on the administrative boundary of the two German states Berlin and Brandenburg, which means that some of the relevant information is only available on one side of the lake.

## 5.2 Vegetation increase

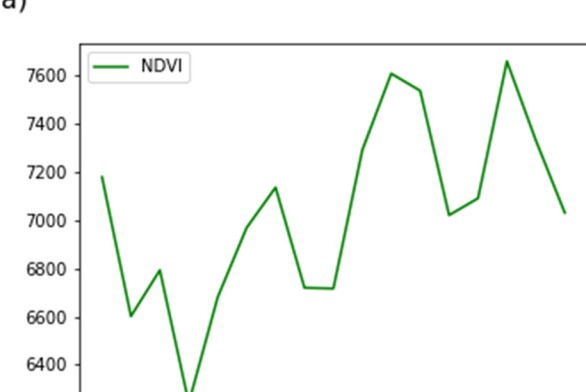

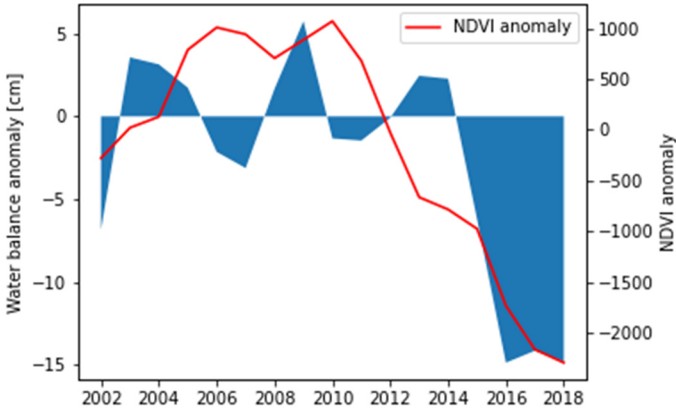

Figure 10: Analysis of vegetation trends: a) yearly average NDVI values integrated over the catchment, b) comparison of the yearly non-climatic water balance anomaly and the cumulative NDVI anomaly of the catchment. The NDVI anomaly is calculated relative to the average NDVI of the 2002-2015 period, before the expected turning point.

A significant part of the lake catchment area west of the lake is covered by forests and heathlands (Fig. 1a). Satellite imagery reveals an increasing trend in the Normalized Difference Vegetation Index (NDVI) between 2002 and 2018 (Fig. 10a). This suggest a general increase in vegetation over the past decades within the catchment area, likely attributed to an increase in forest canopy density and an expansion of woody-vegetation.

Comparing this data with the water balance gap shows striking similarities. In Fig. 10b we calculated the cumulative sum of the NDVI anomaly relative to the 2002-2015 period average, and obtained a similar trend to the water balance anomaly. This suggests a possible connection between the two trends.

A denser canopy intercepts more rainfall available for evaporation, and more mature trees have higher transpiration rates, hence a denser canopy reduces groundwater recharge. The model discrepancies in Fig. 8 are most pronounced in the growing season, where the tree canopies are most developed. This analysis supports the hypothesis that the forest in the catchment has a strong effect on the hydrological system.

The land cover analysis also shows that our modeling could be improved if we could account for the heterogeneous land cover in the catchment when calculating evapotranspiration. Beside the observed 10% increase in NDVI, MODIS evapotranspiration data shows 5-15% increase in forest evapotranspiration in the region (see supplementary figure S2). The impact of this change over the lake levels is equivalent to a yearly flux of 800 000 m$^3$.

This amount could partly explain the water balance deficit, and the increase in evapotranspiration would also explain why the two linear models differ most during the growing season. However, to gain a more precise understanding of the effects of

vegetation cover changes, a more detailed process-based analysis would be required, including biophysical modeling of the trees, and detailed modeling of the recharge process.

### 5.3 Regional groundwater trends

Another hypothesis relates the change in lake dynamics to a larger scale, regional groundwater trend. (Lischeid, 2021) analyzed lake and groundwater level timeseries in the region with principal component analysis. The authors concluded that lakes situated on the higher parts of this lowland region are more sensitive to falling water levels than lakes in the valley bottoms because lakes situated higher are prone to losing their direct connection with the groundwater.

The larger region of Brandenburg has a negative climatic water balance, with water flowing in from areas with a positive budget either as groundwater or surface water flow. This system however is currently under stress not only due to climate change, but also due to the reduced flows of the Spree river, which is caused by the closure of open pit mines in the Lausitz region (HABEL et al., 2023). Therefore, over the last decades, in multiple parts of the region decreasing groundwater levels are visible. As a result, multiple lakes that are mainly groundwater fed show similarly decreasing levels (e.g. Groß Seddiner Lake, Großer Wummsee).

The exact effect of groundwater trends cannot be quantified without groundwater modeling. Still some signs of deviation between the lake and groundwater levels can be seen after 2015, however not decisively clearly. We also cannot attribute the seasonality in the differences between the two periods to this explanation.

### 5.4 Environmental tipping point

Ultimately, it seems most plausible that the observed lake level behavior is the results of a combination of the above-mentioned explanations.

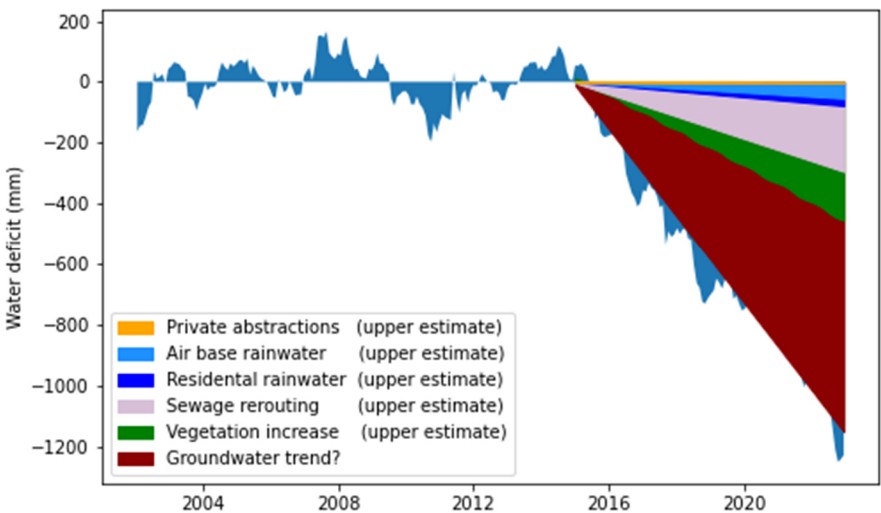

**Figure 11: Potential impacts of the different explanations of the non-climatic water balance anomaly.**

Figure 11 shows a comparison between the estimated impacts of all potential explanations of the non-climatic water balance anomaly considered here. It shows that none of the anthropogenic or ecological factors alone are enough to explain the water deficit completely (even when their maximum potential impact is considered). Note that the effect of regional groundwater trends cannot be estimated, hence we just used it to explain the missing water flux after the other explanations were applied. Based on these results we could not single out one reason that could explain the sudden change in water balance in 2015, but we found multiple processes that probably all contribute to the loss of water. Due to their combined effect, the hydrological system could have reached a tipping point around 2015 that altered the water balance. This tipping point could have been a

critical groundwater level that was reached due to the infrastructural and environmental changes, below which the surface water-groundwater connection got disrupted. The lake bed morphology, or subsurface catchment geometry could be a reason for this critical level to exist.

Another explanation is that the increase in vegetation on the west side of the catchment reduced the groundwater levels locally so that it altered the groundwater flow regime. The gradient of the groundwater table in this area is very small ($3 \times 10^{-4}\ m/m$), hence a local decrease in recharge could divert the groundwater flow and modify the subsurface catchment size. This explanation is in line with our finding that the difference in the hydrological system appears mainly during the growing season. However, to analyze these explanations further, more detailed process-based modeling is required.


## 6    Conclusions

In this paper we have shown how a systematic downward model development approach, using water balance and data-driven models could help with the investigation of a relatively under-studied lake system.

Water balance and data-driven models are well-applicable in such cases, as they mainly rely on observed data which is
generally more available than system knowledge (process understanding). In the current information age, this imbalance is expected to shift even more in favor of data-rich problems. The presented methodology is well transferable to similar groundwater fed lowland lakes, and can be used to identify the major drivers behind the lake level dynamics. The methodology can be adapted for systems with surface water connections, via expanding the models with further features: a net surface water inflow term for the water balance model, and an additional feature for the surface water dynamics in the data-driven model.
With the help of high-resolution weather forcing data and lake level observations we have identified a relation between the climatic and lake level variations. Water balance modeling helped to estimate the inflows and outflows of the system and to reveal any long-term dynamics. Data-driven modeling could then give a more detailed picture of the short-term lake system behavior, including responses to different weather patterns. This set of methods provided an effective toolset for understanding lake level changes and their drivers in a case, where prior hydrological system and process knowledge was limited.
The developed water balance and data-driven models provided very good fits with lake level observation, which shows not just the potential of the modeling approaches, but also the applicability of the CER v2 weather dataset. The approach revealed the main drivers of the lake level dynamics, and provided insight to systemic changes in the hydrological system, which led to hypotheses regarding the lake level loss.

The presented methodology however was not able to clearly identify the exact reason behind the non-climatic lake level loss,
and the proposed hypotheses can only be proved or disproved with additional experiments and/or process-based modeling.
Another drawback of the presented methodology is the strong reliance on good quality data. Closing the water balance, or obtaining a good fit with the linear model was possible only because of the high accuracy of the weather dataset. Due to the spatial variability of precipitation, replacing it with weather station data would lead to a significant drop in model accuracy. Hence in data-scarce regions, robust process-based approaches might be a better solution as they are capable of transferring
knowledge from other comparable catchments, although without data they would operate with large uncertainties.
Our results showed that the lake level variations of the Groß Glienicker Lake between 2002 and 2015 can be explained by the variations in net precipitation, i.e. by precipitation and actual evapotranspiration over the catchment. We have identified a change in the hydraulic system around 2015, which not just resulted in a loss of 4 million cubic meter of water per year, but also changed the hydraulic response of the lake to the climatic inputs.
Increased evapotranspiration from the maturation of a forest in the catchment could explain the altered system dynamics, and the change in the vegetation cover is well aligned with the observed hydrological trend. Therefore, the water loss can be at least partly attributed to the growth of the forest. Another likely reason is the continuously sinking groundwater levels in

Brandenburg due to climate change, which is suggested disrupt the connection of surface water bodies to the groundwater, increasing their outflow. Regional studies show similar lake level trends at several lakes of the area.

Ultimately, the most likely explanation is the combination of the aforementioned processes, which made the hydrological system cross a tipping point during the investigation period. Further analysis of this explanation requires more detailed modeling of the individual processes, and the development of a groundwater model.

The findings of this paper will be used to help the development of such a model. Our main recommendation is the need for a dynamic land cover model that can account for the changes in vegetation and the benefits of gridded meteorological forcing 615 data. While we would still suggest including the water works wells into the model (for which data series are available), we have shown that the effect of private abstractions is negligible, while the effect of infrastructural changes can be significant. Finally, we have emphasized the importance of the regionally observed changes of groundwater levels that need to be considered in any physically-based modelling effort.

**Data availability**

The CER v2 dataset is available from the website of the Chair of Climatology, TU Berlin under the link: https://www.tu.berlin/en/klima/research/regional-climatology/central-europe/cer
The lake level data of the Groß Glienicker Lake can be accessed at the water portal of the Berlin Senate: https://wasserportal.berlin.de/start.php

**Author contributions**

DS, FB and UF prepared and curated the forcing meteorological dataseries. MS did the modelling and the data analysis. AO provided the remote sensing data and did the analysis of the vegetation growth. TK and DS guided the conceptualization and the internal review process. The co-authors all contributed to the preparation of the manuscript.

**Competing interests**

The authors declare no competing interests.

**Acknowledgments**

This study was funded through the Einstein Research Unit 'Climate and Water under Change' from the Einstein Foundation Berlin and Berlin University Alliance under grant number ERU-2020-609.

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
