# Peer review of "A hybrid data-driven approach to analyze the drivers of lake level dynamics"

_EGUsphere, 2023_

## Author Response (AR1)

Dear Dr. Blume,

Thank you for your comments. We have revised our manuscript according to your and the reviewers comments further. Please find some of the specific corrections below.

Going through your responses I find that in quite a few instances you simply answer the reviewers' question without actually explaining how you will incorporate these explanations in the revised manuscript. If the reviewers have trouble understanding and require clarification or additional details this is likely to be the case for future readers as well, so please improve the manuscript accordingly. In some instances, your suggested modifications to the manuscript are very short and might also lack the necessary details. Furthermore, when you have done additional work, such as for example comparing different data sets used for input, it would be good to also provide this information in the manuscript. Some of your suggested changes still require improvement in terms of their phrasing and spelling.

We have expanded our responses, and included further modifications to the manuscript. We have also attached a supplementary figure to show the data used in more detail.

Make sure to provide the units to all equations. To me it seems you have two variables for the change in lake water level Delta S_lake and Delta z. Make sure to be consistent or explain clearly how they differ. Also, clearly differentiate throughout the text when you are talking about lake storage and when about catchment storage. You assume that changes in catchment storage are basically the same as changes in groundwater storage without discussing this further. What is the range in elevation over the catchment and thus the corresponding range in depth to groundwater? How large is your unsaturated zone? Provide a very brief explanation (can also just be one sentence) why this assumption is likely to be correct. Please also include some discussion of the subsurface vs surface catchment issue. You also do not provide sufficient information on the lake water level data you are using. Where was lake water level measured? How was it measured and by whom? This information should be provided in the methods section.

To clarify the differences between delta z and delta S the following text was added:

L211 "The complete water balance then can be used to estimate the changes in catchment storage. To use such model for the lake level dynamics, the catchment storage change ($\Delta S$) needs to be converted to lake level change ($\Delta z$). Lake level change can be estimated from lake storage change using a bathymetric model, but this approach is not suitable for catchment storage. Hence we used an assumption that lake level changes are linearly related to storage changes:"

The elevation range in the catchment is 40 meters, and the unsaturated zone depth is up to 15m (more at the catchment boundaries) according to Karten des Grundwasserflurabstandes Brandenburg 2013. We have included this information to the revised manuscript as:

"The catchment is located in a lowland area, with an elevation range of 40 meters. The unsaturated zone depth is up to 15 meters (Geoportal Brandenburg - Detailansichtdienst 2024).

We added a further comment regarding the unsaturated zone:

L200 "Here, ΔS denotes the change in storage over the whole catchment, which is mainly the change in groundwater storage. With this assumption, the impact of the unsaturated zone is considered within the hydraulic response time. In the model, this time represents the time water need to travel through the unsaturated zone, and it is estimated from the observed data."

Regarding surface and subsurface catchments we added: L302 "The catchment shown in Fig. 1a, was delineated using surface topography data, hence it represents the surface catchment. This is used throughout the analysis, instead of the unknown groundwater catchment. Due to the integrative nature of our analysis, and the focus on the lake level dynamics, this is not a big issue, but it could cause uncertainties when the exact quantification of the hydrological fluxes are needed (see section 4.2)."

Regarding the lake level data source, we have added:

L297 "The lake levels are monitored by an automatic measurement station at the south side of the lake operated by the city of Berlin. Daily lake level data is available since January 1970, and openly available on the website Wasserportal Berlin."

Please also go through your manuscript and check for inconsistencies or imprecision in phrasing. An example from line 192: "Using this formulation, lake levels act as an indicator for groundwater level changes." More precise would be "lake level changes" and not just lake levels. "Using this formulation" is also not entirely clear in its phrasing. Clarify how you are making use of the fact that lake level changes of groundwater dominated lakes are likely to be similar to the changes in the surrounding groundwater. Furthermore, in your response to one of the review comments you state that there is no bathymetric map available for Große Glienecke Lake. This is not true, I did a quick online search and found a bathymetric map in the "Wasseratlas Berlin" from the Berlin Senate.

Thank you! We went through the manuscript to check for such inconsistencies.

Regarding the groundwater connection, we have added:

L177 "In groundwater fed lake systems without any surface water connections, the lake level dynamics will be mainly dependent on the inflow from the groundwater. This flow is controlled by the groundwater head – lake level relation. Hence, the groundwater dynamics and lake level dynamics are strongly related, and the lake level changes can be used as an indicator for the groundwater level changes in the catchment."

There is indeed a bathymetric map available, we have missed that in our search, thank you for the correction. We have included this information into the Study site and data section as:

L299 "The hypsographic curve based on the bathymetric model of the lake shows a linear relation between the lake volumes and lake levels (Jahn and Witt 2002). This suggests that the linear model in equation 4 is a valid assumption, however note that eq. 4 links catchment volume to lake levels, so the hypsographic curve cannot be used directly."
* * *
We would like to thank the reviewers for constructive comments. Please find our replies below the specific comments. Our replies are marked in red, our revisions of the manuscript text are in green.

**Reviewer 1**

This paper about lake level modelling presents an interesting case study with a clearly presented methodology that could readily be transposed to similar cases with little in situ data. The method is overall well presented and the implications for the case study nicely discussed and summarized in the conclusion; the literature review in the introduction is well written; it remains perhaps a little unclear how relevant / important similar case studies are: is a comparable modelling of lake systems a common problem? would most lake systems not be rather different, with numerous hillslopes connected to the lake that require an actual rainfall-runoff modelling part to account for surface inflow? It would perhaps be interesting to check what cited literature refers to similar low land lake systems (without surface water inflow) and which ones to more mountainous / alpine lake systems.

In the area of Brandenburg, East Germany we are aware of at least 5 lakes with no, or very limited surface water connections that are exposed to similar lake level losses: Groß Glienicker Lake, Sacrower Lake, Großer Seddiner Lake, Peetschsee, Straussee. The study of (Lischeid et al. 2021) examines more similar examples from a larger region over East Germany.

Many of these lakes used to have surface water connections, which were dried up due to lake level loss in the last few decades. We think this shows the relevance of such cases, as they will become more and more abundant in the near future.

We modified the study site description as:

L276 "Both lakes are groundwater fed, with no active surface water connections, similarly to multiple lakes in the region (Lischeid 2021). A connection between the two lakes used to exist, but it has been closed in 1996 due to the declining levels in both lakes."

We would also like to note, that the used approaches are not restricted to lakes without surface inflow. Even so, the used data-driven methodology was mainly motivated by data-driven rainfall-runoff modeling studies.

We have also expanded on this in the revised conclusions:

L589 "Water balance and data-driven models are well-applicable in such cases, as they mainly rely on observed data which is generally more available than system knowledge (process understanding). In the current information age, this imbalance is expected to be shift even more in favor for data-rich problems. The presented methodology is well transferable to similar groundwater fed lowland lakes, and can be used to identify the major drivers behind the lake level dynamics. The methodology can be adapted for systems with surface water connections, via expanding the models with further features: net surface water inflow term for the water balance model, and an additional feature for the surface water dynamics in the data-driven model."

Regarding the lakes in more complex environments (such as mountains), we agree that focusing on rainfall-runoff modelling would be the proper methodology, as these lakes are mainly relying on surface runoff. We would still argue that the general downward model development framework (simple to complex) would be still applicable in such cases, and data driven methods could still work under such conditions. This however is beyond the scope of our study.

Detailed comments:

- the balance equations are not well presented; they are a mixture between actual water balance equations and their numerical implementation; please check all units and make sure all quantities have the same units in all equations; do not mix fluxes (in units of volume or mm per unit time) with storage

We have corrected this section. Note that in our equations $\Delta S$ always refers to the total storage change of the respective water body (lake or catchment) as a volume [m³], and P, ET is also taken as volume [m³] integrated over the water body.

L176 "The water balance equation for a groundwater fed lake system can be formulated as:

$$\Delta S_{lake}(t) = P_{lake}(t) - E_{A,lake}(t) + F_{in}(t) - F_{out}(t) + \epsilon$$

( 1 )

where $\Delta S_{lake}$ is the change in lake water storage, $P_{lake}$ is the total precipitation over the lake and $E_{A,lake}$ is the total lake evaporation. $F_{in}$ and $F_{out}$ are in this case the subsurface in- and outflow of water to the lake, which can be combined into the net subsurface water inflow ($\Delta F$). The final term $\epsilon$ explains any remaining errors and uncertainties in the data. If there were any surface water connections to the lake, an extra net surface water inflow would be required to account for it. All terms in eq. 1 are expressed in units of volumes (in m³), with fluxes integrated over the lake surface area.

Precipitation ($P_{catchment}$) and actual evapotranspiration ($ET_{A,catchment}$) over the (groundwater) catchment area, and not just the lake, strongly influence subsurface flow processes that feed the lake. However, these effects show some time delay. Therefore, the water balance equation for the catchment reads as:

$$\Delta S(t) = \int_{t-\tau^*}^{t} P_{catchment}(\tau)\, d\tau - \int_{t-\tau^*}^{t} ET_{A,catchment}(\tau)\, d\tau + \Delta F`(t) + \epsilon$$

( 2 )

"

- "in this study we used a lowpass filter over the lake level data, with a cutoff frequency at 20 days to help with the visualization of the analysis." I would not be able to reproduce the filtering with only this information; filtering comes again later, can you give more indications? is it always clear in the results if the shown data filtered or not?

A zero-phase Butterworth filter was used with a cutoff frequency at 20 days to support the interpretations of the linear regression model.

We thought it interesting that the methods presented in the study were not sensitive to the filtering at all (which is a big contrast compared to machine learning methods for example). Filtering was only relevant for some of the figures relevant to the method illustration. We double checked to make clear where the filtered data was used.

L247 "In this study we used Butterworth filters from the scipy.signal python package. For the autocorrelation analysis in section 4.1 a bandstop filter is used, that removes the 365 days period signal from the lake level data. For the plots of the linear regression analysis (Figure 6,7,8 and 9) a

lowpass filter was used over the lake level data, with a cutoff frequency at 20 days to help with the visualization. A comparison plot of the filtered and unfiltered data is shown in supplementary figure S1."

L312 "A bandstop filter is used over these data to remove the annual cycle, which dominates the lake level periodicity and could distort the analysis (see supplementary figure S1 for the filtered timeseries)."

L364 "The lake level data is filtered using a Butterworth filter with a 20-day cutoff frequency."

- does the system not have surface water in- and outflow? this is not clear in the methods part; it becomes clear in the case study section but this is unusual for most hydrologists, could be made clearer

The system does not have surface in and outflows. There used to be a stream upstream, and a canal connection downstream to the Sacrower Lake, but they both dried out due to the level loss in the last decades.

We have emphasized this property in the methodology section line 182:

"If there were any surface water connections to the lake, an extra net surface water inflow would be required to account for it."

- results: there are details that belong to the methods section, in particular the applied filter to remove the annual signal requires more details; as is, this part is not reproducible; how does the signal look like after the annual signal removal? the part on estimating optimal memory should have a reference

We have added more details to the methods (see our earlier reply). The signal with the annual cycle removed is only used to generate this single figure (Fig. 3a), we have added a supplementary figure to show the modified signal.

L261 "The k-lag autocorrelation shows the time dependence of the lake level data, and give a good indication of the ideal memory timeframe for the modeling (Seeboonruang 2015)."

- testing memory length up to 250 days does not seem to make a lot of sense a priori; why could the system have such a a long memory? what explains the decrease of the metric after 100 days in figure 3?

This was an empirical choice, the length of the timeseries and the small computational costs allowed to analyze longer memories. This was especially relevant during the exploratory phase of our study, when we tested out other methods as well that were more sensitive to memory (such as machine learning approaches). At longer timelags we would be able to see if there are any major periodicities, or maybe the impact of larger scale groundwater systemic effects.

To avoid any confusions, we have replaced the figures with shorter timeframes in the revised manuscript.

L351 "The autocorrelation plot in Fig. 3 shows a rapid decrease, it reaches zero around 20 days, and its minimum value around 30 days. Another method for estimating the optimal memory timeframe in hydrology (mainly in rainfall runoff modeling studies) is to compare linear models with different memory lengths over the same data. This is shown in Fig. 3b, in a range from 1 to 100 days using the $r^2$ metric. The overall picture is very similar to the autocorrelation, with a rapid increase in fit quality until about 20 days. Then the fit quality increase slow down and stays at a high value of 0.8. This large range of optimal fits indicates the robustness and insensitivity of the linear regression method."

[Figure]

- results, line 333: an actual equation of the numerical scheme would be preciser (with correct time steps)

We modified this section as:

L362 "The water balance model is built using eq. 1 on a monthly scale (30 days scale) – suggested by the system memory analysis. The monthly precipitation ($P(t_m)$) and actual evapotranspiration ($ET_A(t_m)$) timeseries are generated via summing up the daily values, and the lake level timeseries (which is used for model validation) is averaged to monthly means. The monthly weather values are then compared with the mean lake level of the next month ($\Delta z(t_{m+1})$).

$$\Delta z(t_{m+1}) = P(t_m) - ET_A(t_m) + \Delta F`$$

( 10 )

"

- line 344: from figure 4, I get that there is too much water in the system, thus line 345 should read that an additional groundwater outflow is required? perhaps I misread the text here, please check

We have corrected this in the revised manuscript.

- line 386: how do you standardize?  the daily time series, how? how can you standardize a time series with many zeros such as precip)?

We used the StandardScaler function from the scikit-learn python package. This is a scaler function modifies the values of the timeseries to approximate a 0 mean normal distribution. As this is just a scaling of the timeseries, the precipitation does not cause any major issues, but the resulted timeseries indeed have a skewed distribution.

Please see:

L250 "To make the linear model coefficients more comparable for the different input types, the input data may be standardized as well. Standardization rescales the input timeseries to have a zero mean and a standard deviation of 1. Standardization may also be necessary for non-linear machine learning approaches."

- line 400: the process description is as if we had a very simple system with a single groundwater system directly connected to the lake; is this the case? very unusual for most hydrologists, for a catchment of 33 km2.

This process description is based on the findings acquired by the presented methodology. These findings can be explained by such simple conceptualization.

However, our current understanding of the site is that the Groß Glienicker Lake has connections to two aquifer layers, that are also connected to each other. There are ongoing studies within the framework of our research project that is focusing on the understanding of the complex groundwater system.

Our presented approach tried to show that to understand the lake level dynamics such complex understanding is not required. The here presented simple process describes the behavior of the simplified model we used. The results show that the majority of the dynamics can be explained with this simplified setup, as shown by the good model fits. The discrepancies between the model and the observation that provided the basis of our discussions shows that the catchment is indeed not as simple as it may seem by the model.

We have modified this section as:

L437 "These findings can be explained with the following conceptualization: after rainfall, as rainwater reaches the groundwater table it creates a hydraulic gradient, and the hydraulic signal reaches the lake very rapidly. An event will affect the water table until the water seeps through the unsaturated soil."

We also would like to refer to the modified conclusions:

L598 "This set of methods provided an effective toolset for understanding lake level changes and their drivers in a case, where prior hydrological system and process knowledge was limited."

**Reviewer 2**

Basically, I find the paper interesting and also agree with the conclusions. There are a couple of good ideas, but also a few technical aspects that should be critically discussed in the revised version of the manuscript. I also do not entirely agree with the strict separation between simple balance models

and complex process-based hydrological models. And that the process-based model is the better choice if sufficient data is available. It always depends... If the components of a simple water balance have been robustly determined (which can often be a lot of work and requires a lot of data) this is not per se less good than a complex process-based hydrological model (with its very own weaknesses and uncertainties such as non-unique solutions etc.).

Thank you for your comments.

Yes, we agree that many of these arguments need to be evaluated critically in a case-per-case basis, but we believe that the general framework (gradual development of models simple to complex) is applicable in general.

We would not draw the line between water balance and process-based methods, instead between data-driven and process-based (because methodologically more complex non-linear methods could be applied within the presented approach, such as machine learning methods).

In our argumentation we would like to differentiate between 'observed data' and 'process data/knowledge'. In the presented case 'observed data' was easily obtainable and available, while knowledge about the hydrological system/process understanding was more limited. We found this a typical scenario of cases that are recently becoming of interest. Under such conditions data-driven techniques could rely solely on the available data.

We would also like to add, that modeling the system dynamics not necessarily requires the estimation of all water balance components. The presented linear regression approach only used the statistical relation between the climate forcing data and the lake levels. A process-based model in the same example would required knowledge about the geology of the catchment (K-values, hydraulic gradient) to make any estimations.

We would also like to emphasize, that our methodology, and the downward model development in general increases the model complexity gradually, which eventually leads into the transition to process-based models, without these being the necessarily better modelling end points. In the presented case, our findings are used in the development of a groundwater model of the same catchment. Hence, we would rather argue that water balance and data driven approaches are ideal starting steps, and great supporting tools within a larger modeling campaign.

L47 "The downward modeling can also fit organically into the development of process-based models, as will be shown."

We did further modifications in the revised manuscript according to our response, please see them at the specific comments.

Line-specific comments:

Line 13: Why „indirectly"?

We mean here the climate-lake level relation. In this formulation, groundwater trends, changing ecosystems and water use are more directly exposed to climatic variations, which then translated to lake level change. We modified this sentence as:

"Lakes are directly exposed to climate variations, as their recharge processes are driven by precipitation and evapotranspiration, but they are also affected indirectly via groundwater trends, changing ecosystems and changing water use."

Line 38: Here you start to explain what you want to do in this study. I would rather put this at the end of the introduction.

We have chosen to keep this section here as it gives an outline of the subsequent sections on the increasingly complex modeling methods, and it also serves as a good introduction to the downward model development.

Line 38: Citation style… only the year should be in brackets. This also applies to various other places in the manuscript.

This is to be adjusted in the final manuscript.

Line 47: Not clear to me what is meant by "higher resolution models of the catchment"

Corrected as:

"higher temporal resolution, or spatially distributed models of the catchment."

Line 51-52: I do not agree with this statement. These handful of hydrological variables/flows are often quite difficult to determine/estimate. Therefore, I would not consider this a typical application in data-scarce regions. As I understand it, process-based models are more likely to be used in data-scarce areas to bridge the gaps of data scarcity (e.g. using standard parameter sets or those from comparable catchments + meteorological forcing data, which is usually quite accessible) – I am not very strict with my opinion here - it should rather be understood as a counter-argument.

Thank you for this comment.

We would still argue that process-based models are more "realistic" in theory but still approximate. In practice, the data requirements to run, parameterize and test them are so high that we have large uncertainties (that we often don't quantify). In such situations the theoretical superiority can be overwhelmed by increased uncertainty. This applies even more so for limited-knowledge regions; if we were honest in these applications then we would have to use very wide prior parameter distributions etc. On the other hand, data-based models often work satisfactorily, especially in forecasting, but maybe for the "wrong" reasons. So best, arguably, is a simplified process-based model that relies on effective parameters estimated from data, traditionally called "conceptual model" in hydrology (despite the ambiguous meaning of the term).

In the revised manuscript, we have replaced the term data-scarce with limited-knowledge, as we find it more fitting based on your comment.

We have also added a critical sentence on our methodology from this perspective to the conclusions:

L609 "Hence in data-scarce regions, robust process-based approaches might be a better solution as they are capable of transferring knowledge from other comparable catchments, although without data they would operate with large uncertainties."

Line 100: Should be "used".

Corrected.

Line 102: What do you mean by "efficient"? I find the word somewhat unsuitable here and would rather write something like "easy to use".

Revised accordingly.

Line 121: Maybe add something like "cannot yet be set up with the required level of confidence".

Revised accordingly.

Line 136: "physics" is a somewhat controversial term in this regard, I would just write "process-based".

Revised accordingly.

Line 149 and line 152: What do you mean with "we propose"? If this is part of your study, just delete "propose". If you propose this for future research (based on the findings of your study), it rather belongs to a conclusion section.

This is for this study, we have corrected it accordingly.

Line 162: Why have you used ERA5 data and not data from the DWD weather station in Potsdam. This data should at least be compared to each other (for validation of the ERA5 dataset). Yes, ERA5 provides actET (as various other products as well), however, how reliable is that? In my experience, not very reliable… At best, this should be compared with the nearest station data (lysimeter) or at least discussed critically (perhaps this is not so relevant because the statistical analysis does not require very precise data… anyway requires some discussion for my taste).

We do not use the ERA5 data directly, but a dynamically downscaled dataset of it (CER v2) specifically created for the Berlin-Brandenburg region. The CER v2 dataset has a higher spatial resolution (2 km spacing) and was validated against 211 DWD weather stations, including the Potsdam DWD station.

We have run all our scripts using weather data from the Potsdam station, but could not achieve the same model fit quality as with the CER data.

The CER v2s dataset was especially better during extreme rainfall events, where the spatial variation in the amount of rainfall can be very large. Because summer storms have very significant effects on the lake levels, it was not possible to close the water balance using DWD station data only.

We were able to recreate the actual evapotranspiration dynamics using DWD station data as well. Here the biggest difference was that we were able to consider the actual land cover composition over the catchment using the gridded CER v2 dataset.

Please see our edits:

L161 "As its predecessor CER v1, the CER v2 dataset has been produced by an observation-based model approach. Global ERA5 reanalysis data has been dynamically downscaled using the Weather Research & Forecasting (WRF) model, and validated against 211 weather stations."

And

L169 "There are two advantages of using a dynamically downscaled gridded dataset instead of relying on interpolated station data. First, such an approach provides an estimate of actual evapotranspiration for each grid point using land cover, vegetation and soil data, and dynamic data on soil moisture, while station-based observations are typically restricted to potential evapotranspiration (lysimeters or eddy flux towers would be available at only very few locations). Second, this approach explicitly takes account for meso-scale heterogeneity of weather systems, which is of particular importance for precipitation and actual evapotranspiration with high variability at spatial scales of a few kilometers or less."

Line 183: Which catchment area you are talking about? Relevant would be the groundwater catchment (right?), however, I worry that the surface catchment area is meant here. In the case of a relatively flat relief (and low hydraulic gradients) and also the fairly strong influence of groundwater extraction, it is quite likely that the subsurface and surface catchment areas differ.

This is correct. The subsurface catchment could also change in time, as we state later in the conclusions. Due to the lack of knowledge, we had to rely on the surface catchment area when defining our models, but we make this a point of discussion to learn about catchment processes in our case.

Note that the differences in catchments are already included in our models implicitly with the net subsurface inflow term, which sums up all the subsurface in and outflows from the catchment area.

Our assumption is that the dynamic changes that control the lake levels are happening within the defined catchment. The results support this assumption, otherwise the model fits should be significantly worse.

We revised the text as:

L185 "Precipitation ($P_{catchment}$) and actual evapotranspiration ($ET_{A,catchment}$) over the (subsurface) catchment area, and not just the lake, strongly influence subsurface flow processes that feed the lake."

And further edits in the data section:

L292 "The catchment shown in Fig. 1a, was delineated using surface topography data, hence it represents the surface catchment. This is used throughout the analysis, instead of the unknown groundwater catchment. Due to the integrative nature of our analysis, and the focus on the lake level dynamics, this is not a big issue, but it could cause uncertainties when the exact quantification of the hydrological fluxes is needed (see section 4.2)."

Line 201: "lake level changes are linearly related to storage changes" In general, I think this assumption is far too simplistic. I don't think this oversimplification is necessary either, as only a

bathymetric model would be needed to represent this correctly. A quick Google search shows that such a bathymetric model exists and is accessible.

We agree, that in many cases this assumption could be too simple. In the presented example, we think the fit quality of the water balance results show that this assumption was enough. This is due to the very integrative nature of our study, as we are not focusing on any spatial heterogeneities in the catchment or lake.

The bathymetric model of the lake shows a linear relation between the volume and the lake level. However, this model is valid for the water body of the lake and not the catchment. There this relation might be different.

We added the following:

"The hypsographic curve based on the bathymetric model of the lake shows a linear relation between the lake volumes and lake levels (Jahn and Witt 2002). This suggests that the linear model in equation 4 is a valid assumption, however note that eq. 4 links catchment volume to lake levels, so the hypsographic curve cannot be used directly."

Line 209: The application of the strong oversimplification, mentioned in the previous comment, quite likely has a systematic impact/error on the estimation of dF and thus the identification of tipping points (e.g., presumed tipping points might actually be related to lake bed morphology).

Interesting, thank you. We added this argument as a possible explanation for the tipping points in our discussion section.

L582 "This tipping point could have been a critical groundwater level that was reached due to the infrastructural and environmental changes, below which the surface water-groundwater connection got disrupted. The lake bed morphology, or subsurface catchment geometry could be a reason for this critical level to exist."

With regard to the two previous comments, you could possibly also argue that in your particular case the water level decrease only takes place in a relatively small range and therefore such a linear assumption is not completely wrong... (but should be checked with the bathymetric model)

Thank you for this comment. We think this might be an argument for the generalization of the method, as this assumption would hold up in many lake or catchment geometries.

L212 "Focusing on water level changes in a limited range the linear assumption should be a good approximation, but note that under complex hydrological conditions this relation may be different. Still, this would be seen if the observed and modelled timeseries would not come close during this optimization."

Line 228: Delete the word "same" or replace it by "respective".

Revised accordingly.

Line 239: I not really get why filtering helps neural networks? Such a statement should be explained.

Removing some of the higher-frequencies, remove some of the non-linearity of the problem. This makes the calibration of high-parameter models less ill-posed, hence more robust.

In this paper filtering was necessary for visualization, hence we modified this section to avoid any confusion as:

L246 "Optionally, input data might be filtered prior to the analysis. In this study we used Butterworth filters from the scipy.signal python package. For the autocorrelation analysis in section 4.1 a bandstop filter was used, that removes the 365 days period signal from the lake level data. For the plots of the linear regression analysis (Figure 6,7,8 and 9) a lowpass filter was used over the lake level data, with a cutoff frequency at 20 days to help with the visualization."

Line 278: What exactly do you mean with "civil engineering information"? Please, specify.

Revised as:

"and limited information civil engineering informationon the rainwater infrastructure (e.g. manhole cover locations) is available."

Line 300-306: This belongs to the results section.

We moved these lines to the results together with lines 285-290 to the results.

Line 308-310: These lines are superficial.

We deleted these lines.

Line 324: Isn't that actually a pretty vague result? Couldn't it be that the memory size might not be larger, e.g. 100 days? How would the model results change if a different memory size were assumed?

Originally Figure 3b showed that a very wide range of memory sizes are suitable for modeling. We have checked, and the modelled results of the linear model are very similar, even when using a suboptimal memory of 150-200 days. This is due to the fact that the model coefficients of higher days are small compared to the days closer to t=0 (see Fig. 7).

We have noted this insensitivity in the revised manuscript:

L355 "The large range of optimal fits indicates the robustness and insensitivity of the linear regression method."

After the recommendation of the first reviewer, we have reduced the presented memory ranges on the figure.

Line 325: "One might link this time to the catchment size, as the distance travelled by the groundwater flow." I would also delete this sentence. (It's not a good style to assume that that the reader doesn't know the very obvious fact that this is caused by pressure transmission…)

Revised accordingly.

Line 415-427 and Figure 8: How representative are the selected time periods in 2006 and 2016? Is it valid to draw general conclusions from them (they could simply be singular phenomena…)? Either this should be well argued in the text or average values could simply be used as in Figure 9.

Although we present only single events, we have looked through the whole timeperiod to draw our conclusions (and we found these examples to be representative). These two periods were good

examples, where the differences in the model behaviors can be discussed in detail. We found this more hands on, than only showing the averaging of Fig. 9.

Averaging could not show the impact of single rainfall events, or drought periods, as these events happen different times in different years, hence we find it important to show this also.

We expanded as:

L453 "In this plot we zoom into two different parts of the dataset to compare the two models directly in detail, focusing on typical weather events."

Line 444: Instead of "since 2015", you should provide the exact time period, i.e. "between 2015 and XXX". This also applies to later cases in the text, e.g. line 451.

We have checked and corrected all instances to "between 2015 and 2022".

Line 461: Why only 4%? Even if one takes 50% as the average (which could definitely be regarded as the upper limit), the influence would only be marginal.

4 % was taken from an online draft version of the paper (Schleich and Hillenbrand 2009), but the actual published number is 7% (as an upper bound).

We have corrected it and added the reference to the revised version.

Line 487: "some of the relevant information is only available on one side of the lake" Would be good to be a bit more specific, i.e. which information is missing on which side.

This refers to the Study site and data section where:

L276 "From the Berlin side, data regarding water supply, wastewater management and canalization maps are available on the city webpage for multiple years. From the Brandenburg side, geological and limited information civil engineering information the rainwater infrastructure (e.g. manhole cover locations) is available."

Figure 10: Why you took 2010 as a baseline? And is a baseline necessary? For Figure 10b you could also just sum up the anomalies (e.g. as deviation from the mean) starting at zero. And why are the cumulative anomalies going down when the NDVI increases (and are above the reference/baseline value)? – if I misunderstood something, it should be better explained…

We choose a baseline for visualization purposes. 2010 was chosen as it is at the middle of the timeframe, and its value is close to the mean NDVI of the 2002-2015 period. The mean of the complete timeseries would consider the period after the investigated tipping point, which we wanted to avoid here.

The NDVI anomaly was defined as ($NDVI_{2010}$-NDVI), so larger NDVI makes the anomaly go down to show the increase in water uptake. We have added this definition to the figure caption.

To avoid such confusions, we replaced the baseline, with the 2002-2015 period average. We have also removed it from the first plot for further clarity.

The caption was revised as:

"Figure 10: Analysis of vegetation trends: a) yearly average NDVI values integrated over the catchment, b) comparison of the yearly non-climatic water balance anomaly and the cumulative NDVI anomaly of the catchment. The NDVI anomaly is calculated relative to the average NDVI of the 2002-2015 period, before the expected turning point."

And the text in the main body as:

L536 "In Fig. 10b we calculated the cumulative sum of the NDVI anomaly relative to the 2002-2015 period average, and obtained a similar trend to the water balance anomaly. This suggests a possible connection between the two trends."

Line 504-505: Basically, you are saying that an increase of 10% in NDVI causes an increase of 5-15% in ET. Do you have any reference for this or at least indications for this assumption?

Unfortunately, there is no easy way to link NDVI to ET without using vegetation models. To show an estimate we looked directly at MODIS satellite data from the catchment which showed this increase in ET.

We have rephrased as:

L544 "Beside the observed 10% increase in NDVI, MODIS evapotranspiration data, shows 5-15% increase in forest evapotranspiration in the region."

Line 508: One of the "However" is too many.

Corrected.

Line 512-521: I agree to the hypothesis that the evolution of groundwater levels at a regional scale has an impact (although it doesn't explain the tipping point). However, doesn't that somewhat contradict your model assumption that your subsurface catchment corresponds to your surface catchment?

Please see our earlier response to the catchment definition again.

Regional groundwater trends could impact the subsurface inflow to the catchment, which could lead to lake level loss. The tipping point can happen if the connection between the lake and the groundwater is lost.

L552 "(Lischeid, 2021) analyzed lake and groundwater level timeseries in the region with principal component analysis. The author's concluded that lakes situated on the higher parts of this lowland region are more sensitive to falling water levels than lakes in the valley bottoms because lakes situated higher are prone to losing their direct connection with the groundwater."

The groundwater flow directions could also change, that could lead to a tipping point, similarly as discussed for the vegetation in L568:

"Another explanation is that the increase in vegetation on the west side of the catchment reduced the groundwater levels locally so that it altered the groundwater flow regime. The gradient of the groundwater table in this area is very small ($3 \times 10^{-4}\ m/m$), hence a local decrease in recharge could divert the groundwater flow and modify the subsurface catchment size."

Line 540-541: Good point! But, again, it somehow contradicts your model assumption as described in the previous comment.

See our previous reply.

Line 546-570: This is rather a summary of your results than a conclusion of your study. I would more focus on pros and cons of your study (maybe also some of my comments/critics can be discussed here).

Thank you. We have extended our conclusions in the revised manuscript according to your suggestion. We have also trimmed down some of the summary to further streamline the text (deleted lines 557-561).

L598 "This set of methods provided an effective toolset for understanding lake level changes and their drivers in a case, where prior hydrological system and process knowledge was limited.

The developed water balance and data-driven models provided very good fits with lake level observation, which shows not just the potential of the modeling approaches, but also the applicability of the CER v2 weather dataset. The approach revealed the main drivers of the lake level dynamics, and provided some insight to systemic changes in the hydrological system, which led to some possible hypotheses regarding the lake level loss.

The presented methodology however was not able to clearly identify the exact reason behind the non-climatic lake level loss, and the proposed hypotheses can only be proved or disproved with additional experiments and/or process-based modeling.

Another drawback of the presented methodology is the strong reliance on good quality data. Closing the water balance, or obtaining a good fit with the linear model was possible only, because of the high accuracy of the weather dataset. Due to the spatial variability of precipitation, replacing it with weather station data would lead to a significant drop in model accuracy. Hence in data-scarce regions, robust process-based approaches might be a better solution as they are capable of transferring knowledge from other comparable catchments, although without data they would operate with large uncertainties. "

**Publication bibliography**

Geoportal Brandenburg - Detailansichtdienst (2024). Available online at https://geoportal.brandenburg.de/detailansichtdienst/render?url=https://geoportal.brandenburg.de/gs-json/xml?fileid=A140C263-7D61-447B-81C2-8824792AE190, updated on 4/29/2024, checked on 4/29/2024.

Jahn, Dietrich; Witt, Henner (2002): Gewässeratlas von Berlin. Senatsverwaltung für Standentwicklung. UNICOM, Berlin. Download:< http://www. berlin. de/sen/umwelt/wasser/wasserrecht/pdf/wasseratlas. pdf.

Lischeid, Gunnar (2021): Abschätzung des mittelfristigen Niedrigwasserrisikos anhand der Daten des Grundwassermonitorings. In KW Korrespondenz Wasserwirtschaft (12), pp. 780–785. DOI: 10.3243/kwe2021.12.004.

Lischeid, Gunnar; Dannowski, Ralf; Kaiser, Knut; Nützmann, Gunnar; Steidl, Jörg; Stüve, Peter (2021): Inconsistent hydrological trends do not necessarily imply spatially heterogeneous drivers. In *Journal of Hydrology* 596, p. 126096. DOI: 10.1016/j.jhydrol.2021.126096.

Schleich, Joachim; Hillenbrand, Thomas (2009): Determinants of residential water demand in Germany. In *Ecological Economics* 68 (6), pp. 1756–1769. DOI: 10.1016/j.ecolecon.2008.11.012.

Seeboonruang, Uma (2015): An application of time-lag regression technique for assessment of groundwater fluctuations in a regulated river basin: a case study in Northeastern Thailand. In *Environ Earth Sci* 73 (10), pp. 6511–6523. DOI: 10.1007/s12665-014-3872-7.

---

## Author Response (AR2)

**Public justification (visible to the public if the article is accepted and published)**:
Dear Authors,

both reviewers have studied your revised manuscript and are happy with the changes you made. However, there are still some corrections to be made according to Review #2. Furthermore, Reviewer #2 also comments on errors in terms of language. As I also found several mistakes and your phrasing sometimes too vague or awkward and thus not clear for the reader, please again take the time to go over your writing. I have added some more comments and suggestions in the attached pdf (inserted into your response, but of course to be applied to the manuscript).
I am recommending minor revisions and as the required changes are indeed not major, I am not planning to send the manuscript back out to the reviewers.

Looking forward to receiving your revised manuscript and concluding the publication process!
All the best,
Theresa Blume

Thank you for the comments and suggestions! Please find our corrections below.

- Black text denotes the review comments.
- Green color denotes original text from our previous document.
- Red is our responses and corrections.

The revised version addresses all my previous comments. But the modified text contains several langues mistakes, to be corrected.

I have some minor suggestions:

- do not use groundwater level and groundwater head interchangeably, use precise wording
- line 189: with fluxes integrated over the lake surface area "and in time".

Corrected

- line 200: "With this assumption, the impact of the unsaturated zone is considered within the hydraulic response time": I do not understand what this means; do you not simply mean that storage changes in the unsaturated area are neglected?
- line 200: " In the model, this time represents the time water need spends travelling through the unsaturated zone, and it is estimated from the observed data"; please avoid giving the impression that hydraulic memory is related to actual water travelling through the system; you rather refer to the time that the impulse is travelling through the system; this difference is important (if needed, refer to the transit time estimation work by James Kirchner for details)

Corrected as:

"With this assumption, storage changes in the unsaturated zone are neglected. In the model, this time represents the time water spends travelling through the unsaturated zone, and then the pressure impulse traveling through the system. The hydraulic memory of the system is estimated from the observed data."

-bandstop filter: can you give the implementation also as for the other filter?

Both implementations are included in the supplementary figure.

- fig. 3b: linear models - what is modeled and what enters the linear model (should be clear already here)

We rephrased the caption as:

"fit quality of linear models of the weather forcing-lake level relations (eq. 6) using different system memory timeframes."

Editor comments

L211

please always state precisely if you are referring to lake or catchment storage. This needs to be clarified. Please also explain why you assume that. What is the basis of this assumption? Right now the phrasing makes it sound a bit random.

"We based this assumption on the fact that lake level changes are relatively small compared to the scale of the catchment, and the catchment geometries are simple in a lowland, sedimentary geological setting. The catchment storage change - lake level change relation reads as:"

Fig. 3

is memory size the correct term here for memory extent in days? It sounds awkward. Maybe system memory?

For more clarity, we removed the title from the top of the plots, and rewrote the caption as:

"Lake response time analysis: a) autocorrelation of lake levels, b) fit quality of linear models using different system memory timeframes."

L400

Our presented approach tried to show that to understand the lake level dynamics such complex understanding is not required. The here presented simple process describes the behavior of the simplified model we used. The results show that the majority of the dynamics can be explained with this simplified setup, as shown by the good model fits. The discrepancies between the model and the observation that provided the basis of our discussions shows that the catchment is indeed not as simple as it may seem by the model.

this seems a bit circular but also contradictory,

We would revise the last sentence as: The analysis of the remaining discrepancies between the model and the observation that provided the basis of our discussions shows that the catchment is not as simple as it may seem by the model.

L437

An event will affect the water table until the water seeps through the unsaturated soil.

I don't understand this

Corrected as:

"A rainfall event will affect the water table until the water seeps through the unsaturated soil."

**Reviewer 2**

"Lakes are directly exposed to climate variations, as their recharge processes are driven by precipitation and evapotranspiration, but they are also affected indirectly via groundwater trends, changing ecosystems and changing water use."

I find this separation of directly and indirectly a bit hard to follow. If the lake is groundwater fed and groundwater recharge is affected by climate change - that is still a direct impact to me. You also also seem to consider impacts on the recharge processes as direct impacts. Clarify why groundwater trends caused by climate change are not a direct impact. If you are here referring to groundwater trends caused by other processes then this should be clarified.

We rephrased this, to avoid confusion:

"Lakes are directly exposed to climate variations, as their recharge processes are driven by precipitation and evapotranspiration, and they are also affected by groundwater trends, changing ecosystems and changing water use."

Comment to L162:

It would be good to mention this also in the manuscript, I already suggested this in the previous round.

Included as:

"Second, this approach explicitly takes into account meso-scale heterogeneity of weather systems, which is of particular importance for precipitation and actual evapotranspiration with high variability at spatial scales of a few kilometers or less. When we tested our lake models using weather station data, we were unable to obtain the same model fit qualities as with the CER v2 dataset. The largest differences happened after extreme rainfall events, where due to the spatial variation the recorded amount of rainfall could differ a lot from the rainfall at other locations. Because summer storms have a strong impact on the lake levels, we could not close the water balance models using weather station data."

L201

This suggests that the linear model in equation 4 is a valid assumption, however note that eq. 4 links catchment volume to lake levels, so the hypsographic curve cannot be used directly."

You need to explain why this is a valid assumption when you are using catchment storage and not lake storage. Why is it possible to assume that if your lake shows a linear relationship, the catchment will also do that? When is this the case and when isn't it?

The lake level – catchment storage relation is mainly a question of geological complexity. The lakebed morphology could be an indicator of a complex, or simpler geological setting, hence linearity in the hypsographic curve could be an indicator as well. But, as many other factors would affect this relation as well, after some consideration we decided to remove this argument from the revised manuscript.

Instead, for our choice of linearity we would use the argument of the assumed simple geology directly, and the argument of scale differences raised by the second reviewer: as the range of lake level changes are much smaller then the catchment scale. Please see our first correction above.

We rewrote this section as:

"The hypsographic curve based on the bathymetric model of the lake shows a linear relation between the lake volumes and lake levels (Jahn and Witt, 2002), but note that this relation cannot be used to link the lake levels to the catchment storage."

L212 – we have removed this sentence.

L246 - in what way does this help with visualization? Needs to be clarified.

"For the plots of the linear regression analysis (Figure 6,7,8 and 9), a lowpass filter was used over the lake level data, with a cutoff frequency at 20 days. This was necessary for the visualization in Fig. 7, where the higher frequency components would appear as noise over the coefficients."

L324

"larger lags"

L544 – We show now the MODIS data in supplementary figure 2.

---

## Author Response (AR3)

Dear Dr. Blume,

Thank you for your comment.

What we meant with the second sentence is, that after rainfall not all water seeps through the soil at once, but some of it takes more time. Indeed, these slower components are affected by other processes as well, but in the model it still shows as a continuous slow decay of the precipitation coefficients.
We rewrote the section to explain this better:

"These findings can be explained with the following conceptualization: after rainfall, as rainwater reaches the groundwater table it creates a hydraulic gradient, and the hydraulic signal reaches the lake very rapidly. The impact of the rainfall is still visible a few days later, as some of the water takes more time to seep through the soil. This impact decays over time continuously."